# Improving deep learning-based segmentation of diatoms in gigapixel-sized virtual slides by object-based tile positioning and object integrity constraint

Michael Kloster[1]*, Andrea M. Burfeid-Castellanos[1], Daniel Langenkämper[2], Tim W. Nattkemper[2], Bánk Beszteri[1]

**1** Department of Phycology, Faculty of Biology, University of Duisburg-Essen, Essen, Germany, **2** Biodata Mining Group, Faculty of Technology, Bielefeld University, Bielefeld, Germany

* michael.kloster@uni-due.de

**Data Availability Statement:** Supplemental data is available from the Zenodo repository: S1 Dataset. Virtual slide images of diatom preparations from

## Abstract

Diatoms represent one of the morphologically and taxonomically most diverse groups of microscopic eukaryotes. Light microscopy-based taxonomic identification and enumeration of frustules, the silica shells of these microalgae, is broadly used in aquatic ecology and bio-monitoring. One key step in emerging digital variants of such investigations is segmentation, a task that has been addressed before, but usually in manually captured megapixel-sized images of individual diatom cells with a mostly clean background. In this paper, we applied deep learning-based segmentation methods to gigapixel-sized, high-resolution scans of diatom slides with a realistically cluttered background. This setup requires large slide scans to be subdivided into small images (tiles) to apply a segmentation model to them. This subdivision (tiling), when done using a sliding window approach, often leads to cropping relevant objects at the boundaries of individual tiles. We hypothesized that in the case of diatom analysis, reducing the amount of such cropped objects in the training data can improve segmentation performance by allowing for a better discrimination of relevant, intact frustules or valves from small diatom fragments, which are considered irrelevant when counting diatoms. We tested this hypothesis by comparing a standard sliding window / fixed-stride tiling approach with two new approaches we term object-based tile positioning with and without object integrity constraint. With all three tiling approaches, we trained Mask-R-CNN and U-Net models with different amounts of training data and compared their performance. Object-based tiling with object integrity constraint led to an improvement in pixel-based precision by 12–17 percentage points without substantially impairing recall when compared with standard sliding window tiling. We thus propose that training segmentation models with object-based tiling schemes can improve diatom segmentation from large gigapixel-sized images but could potentially also be relevant for other image domains.

river Menne (https://doi.org/10.5281/zenodo.7078938). S2 Dataset. Python scripts / Jupyter Notebooks and data for training segmentation models on virtual slide images of diatom preparations from river Menne (https://doi.org/10.5281/zenodo.7079005). S3 Dataset. Mask R-CNN and U-Net segmentation models and experimental results on segmenting virtual slide images of diatom preparations from river Menne (https://doi.org/10.5281/zenodo.7079072). S4 Dataset. Segmentation performance evaluation results and R scripts for generating Figs 5 and 6 as well as Tables 3 and 4 (https://doi.org/10.5281/zenodo.7107456).

**Funding:** This work was funded by the Deutsche Forschungsgemeinschaft (DFG) in the framework of the priority programme SPP 1991 Taxon-OMICS under grant nrs. BE4316/7-1 & NA 731/9-1. D.L.'s contribution was supported by the German Federal Ministry for Economic Affairs and Energy (BMWi) (FKZ: 0324254D). BIIGLE is supported by the BMBF-funded de.NBI Cloud within the German Network for Bioinformatics Infrastructure (de.NBI) (031A537B, 031A533A, 031A538A, 031A533B, 031A535A, 031A537C, 031A534A, 031A532B). The funders had no role in study design, data collection and analysis, decision to publish, or preparation of the manuscript.

**Competing interests:** The authors declare no competing interests.

## Introduction

Diatoms are one of the morphologically and taxonomically most diverse groups of microscopic eukaryotes, belonging to the Stramenopiles, and possessing uniquely fine-structured silicate shells called frustules [1, 2]. Taxonomic identification and enumeration of diatoms by light microscopic observation of their frustules and valves, which are the terminal plates of the frustules, is broadly used in aquatic ecology and biomonitoring [3]. There is a recognized need for reducing subjectivity and increasing the reproducibility of light microscopic taxonomic identification of diatoms, as well as for speeding up routine analyses. This has led to extensive research on digitalization and automation of the light microscopy workflow [4, 5]. The main methodological steps in this context include large-scale automated light microscopic image acquisition at high optical resolution [6–12]; localization of diatom frustules / valves in images [13–19]; extraction of morphometric descriptors from image patches depicting diatom frustules / valves [12, 13, 20–30]; and taxonomic identification, i.e., labelling of such image patches with a taxonomic name [26, 31–33].

One of the most promising approaches towards large-scale light microscopic image acquisition from standard diatom microscopy slides is high-resolution slide scanning. In early proposed diatom slide scanning procedures, a first scanning step was performed with a low-resolution (air) objective, and the resulting images were only used to localize diatom valves using algorithmic approaches [6]. These valves were then re-visited for a second imaging, this time with a high-resolution oil immersion objective giving sufficient detail for taxonomic identification of the concerned diatom [10]. However, the re-visiting step only became feasible with the advent of high-precision commercial slide scanning microscopes. More recently, it has also become possible to directly perform and process high-resolution scans of large continuous slide areas, leading to substantially simplified slide scanning protocols. Here, a section of a microscopy slide, often several dozens of square millimeters, is covered entirely by large numbers of overlapping field-of-view images, which subsequently are stitched together. This results in images up to several gigapixel in size, which are often termed virtual slides, a terminology we will also use in this paper. A similar procedure is established in pathology as whole slide imaging or WSI [34], although the use of high-resolution oil immersion objectives, as necessary for diatoms, is rather rare there. Availability of software tools to work above a certain image size limit (2 GB) has, however, been limited, making work with such slide scans highly challenging until recently. Nevertheless, many of these challenges are being overcome, so that it is now becoming possible to apply these methods for manual or software-assisted annotation [35], for instance for morphometric studies or generating training image sets for deep learning classification models [29, 33].

Presently, a critical gap in digital / automated diatom analysis using high-resolution virtual slides is the step of locating individual diatom frustules or valves in gigapixel-sized images. This can be performed in different ways. In recent computer vision literature, object detection, mostly only aiming at delineating a bounding box around objects of interest, is differentiated from segmentation. The latter could be defined as classification of individual pixels in an image into two or more categories, usually background vs. one or more object classes. Instance segmentation is sometimes cast as a combination of object detection and segmentation, in that a pixel-accurate segmentation of individual objects is attempted. This last concept corresponds best to previous work aimed at locating diatoms in digital images, although much of that works preceded the current terminology [13–17]. Thus far, however, these methods have not been applied to gigapixel-sized virtual slide images, but usually to images taken manually and centering on individual diatom frustules. Ruiz-Santaquiteria et al. [18] utilized semantic segmentation on individual field-of-view images containing multiple diatoms. These images were

processed at a stretch, but were scaled down to the substantially smaller input size of the segmentation models, thus loosing resolution in the generated segmentation masks.

In one study, diatom localization in slide scans of medium resolution, obtained with dry objectives, was attempted without applying object detection or segmentation methods. Instead, Zhou, Zhang [11] applied a classifier model to small subsections from a slide scan, outputting a decision whether a diatom can be found in a specific subsection or not.

Gigapixel-sized images are too large to be directly fed into typical deep learning-based segmentation models such as Mask R-CNN [36] or U-Net [37] at original resolution. A common solution is to break such large images down into smaller subsections fitting the model's input size, often referred to as tiling. Tile positions are often arranged in a "fixed-stride" pattern so that adjacent tiles overlap by a certain, fixed margin [38], also referred to as sliding window approach. The segmentation model predictions of all tiles are then combined with respect to the original position of the tile, ultimately resulting in a segmented version of the complete gigapixel image. This process often is referred to as stitching.

A problem arising from tiling the image data is that intact / complete objects can be cropped randomly by a tile's boundaries and thus coincidentally resemble small object fragments. However, diatom fragments comprising less than roughly 75% of the complete valve / frustule usually are rejected in aquatic ecology and biomonitoring investigations [39, 40]. In the following, we will refer to such fragments as irrelevant small fragments and use the term relevant or mostly intact diatom objects for valves or frustules which are at least 75% complete. We hypothesized that applying a tiling scheme that reduces the amount of artificially cropped target objects fed to the networks during training might improve segmentation performance for diatom virtual slides.

To test this hypothesis, we a) used slide scanning to image large sections of diatom microscopy slides in form of gigapixel-sized virtual slide images; b) implemented automated pixel-accurate segmentation of diatoms in these virtual slide images utilizing Mask R-CNN and U-Net models, c) investigated factors influencing segmentation performance and d) developed and assessed two tiling methods that reduced the extent of random cropping in relevant diatom valves / frustules.

## Material and methods

We investigated benthic diatoms sampled at six different locations from the Menne river comprising more than 110 different, morphologically diverse taxa (estimated by identifying a subset of specimens; for the experiments presented below, specimens were only annotated as belonging to the object class "diatom"). Typical representatives of the ten most abundant species are shown in Fig 1, where frustules of some species are oriented in different view angles. The sampling of this stream was authorized by the Kreis Paderborn authorities through the 4032-20-450 permit.

### Sampling and preparation

For each of the six sampling locations, diatoms were sampled using the standard norm [41], scraping 20 cm$^2$ biofilm of 5 stones. The raw material was cleaned following the "hot $H_2O_2$–HCl" protocol [40]. After washing, the sample-dripped coverslips were mounted with Naphrax artificial resin (Thorns Biologie Bedarf, Deggendorf, Germany) as permanent slides. These contained intact diatom valves and frustules as well as valve / frustule fragments, diatom girdle bands and copious amounts of non-diatom particles like silt or clay (Fig 2).

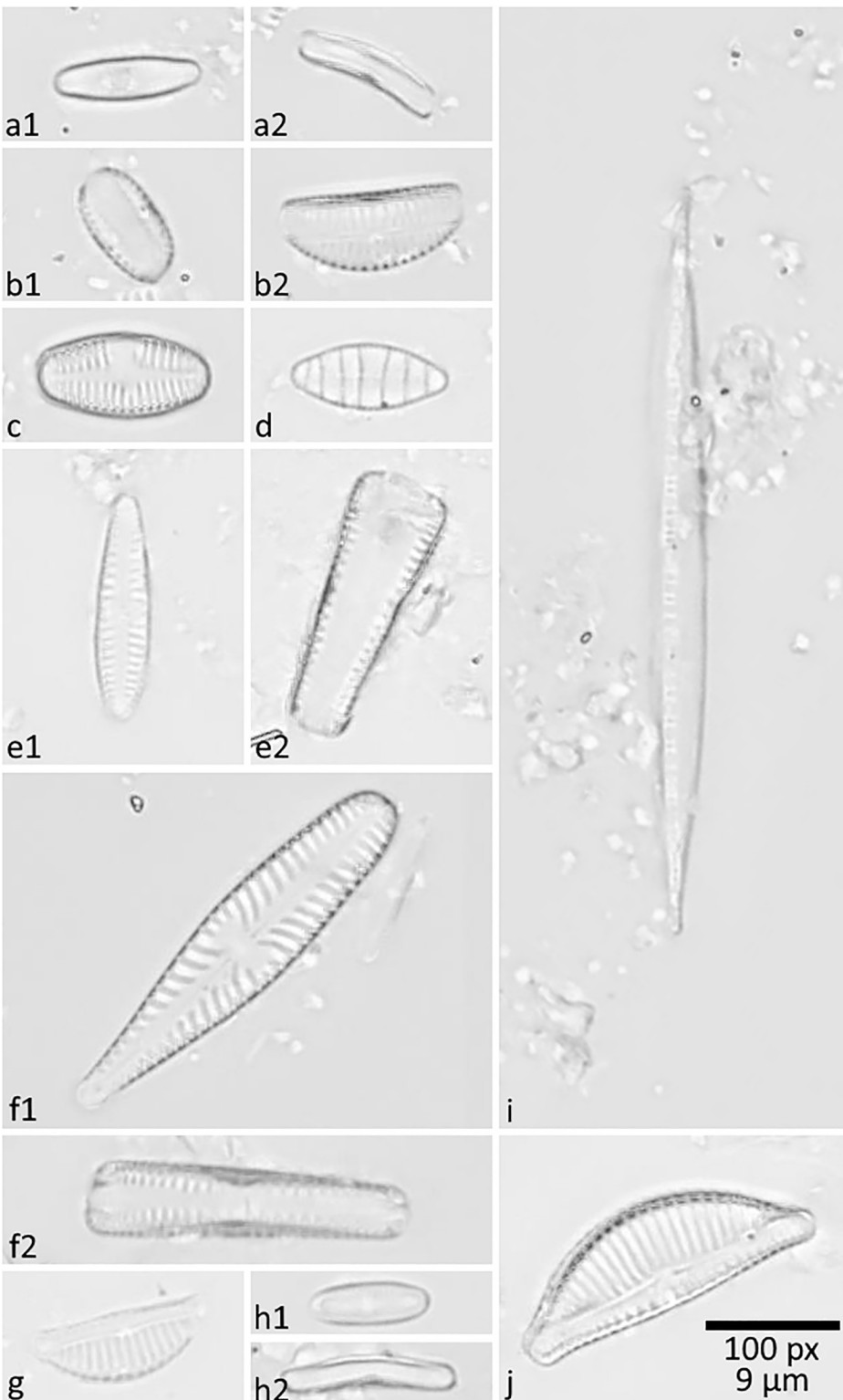

**Fig 1. Typical examples of the ten most abundant diatom species to demonstrate their morphological diversity.**
a1, a2) Achnanthidium minutissimum in valvar and pleural view; b1, b2) Amphora pediculus in valvar and pleural view; c) Planothidium lanceolatum; d) Denticula tenuis; e1, e2) Gomphonema pumilum var. rigidum in valvar and pleural view; f1, f2) Gomphonella olivacea in valvar and pleural view; g) Encyonema minutum; h1, h2) Achnanthidium rivulare in valvar and pleural view; i) Nitzschia dissipata; j) Encyonema ventricosum.

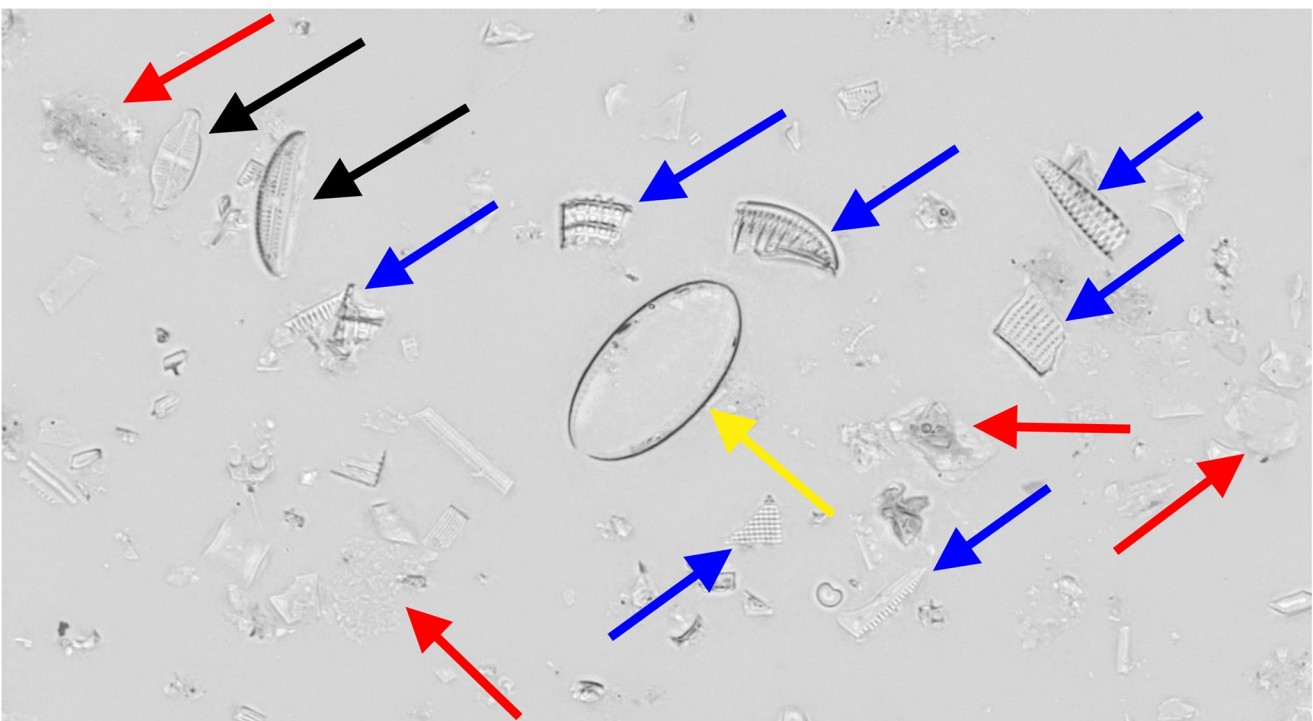

**Fig 2. Small subsection of a virtual slide image, containing relevant, (mostly) intact diatom objects (black arrows) as well as irrelevant small diatom fragments (blue) and girdle bands (yellow) and also non-diatom debris like silt or clay particles (red).**

## Imaging

From each of the six slides an area of approximately $5 \times 3.5$ mm$^2$ was scanned by an VS200 slide scanning microscope (Olympus Europa SE & Co. KG, Hamburg, Germany). 61 focal planes were imaged in bright-field mode at a distance of 0.28 μm each using a UPLXA-PO60XO 60x/1.42 oil immersion objective. Focus stacking and stitching were performed with the built-in functions of the VS200 ASW software (v3.2.1). Subsequently the RGB color data was reduced to 8bit monochrome grayscale images. This process resulted in six virtual slide images, each roughly $55,000 \times 39,000$ pixel (S1 Dataset), with a resolution of ca. 0.09 μm / pixel and covering an extended depth of field of ca. 17 μm. As a side effect of focus stacking, diatom objects which were not perpendicular to the optical axis of the imaging system might appear slightly smaller [10].

## Annotation

In these virtual slide images relevant diatom objects, i.e. valves and frustules which were estimated to be at least about 75% complete by the human annotator, were labelled as class "diatom" by manually defining their rough outline using the BIIGLE web service [42]. From each virtual slide image, one rectangular section was selected so that it contained approximately 500 relevant diatom objects (Fig 3, where this section is comprised by the three rectangles on the top). This area covered the complete width of the virtual slide image (55,000 pixel), but varied in height (1,700–39,000 pixel) depending on the density of diatoms within the sample. Smaller diatom fragments and very dense aggregates of diatom material, where it was not possible to tell apart individual cells, were not considered. Every annotation was seen and checked by at

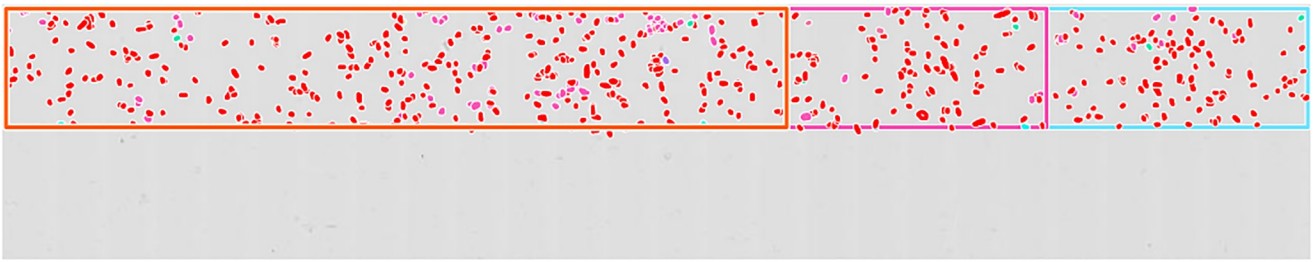

**Fig 3. Screenshot of a vertically cropped area of an annotated virtual slide image.** The original size of the complete virtual slide is approximately 55,000 x 39,000, the size of the depicted clipping 55,000 x 11,000 pixel. In the upper part of the image about 500 diatom objects were annotated (colored spots) and subdivided into training, validation and evaluation subsections (orange, pink and cyan rectangles, respectively).

least two different persons to ensure the annotated object was a relevant diatom and that its outline was captured accurately.

Subsequently, cutouts encompassing each annotation were produced using our software tool SHERPA2BIIGLE and segmented using SHERPA [13] with manual rework to capture the exact outline for each annotated diatom object (Fig 3, colored spots). Any pixel not labelled as class "diatom" was considered as class "background".

## Data partitioning

For each virtual slide image, the annotated area was subdivided into three non-overlapping rectangular subsections covering 60%, 20% and 20% of the area (Fig 3, colored rectangles). From these, data were extracted for segmentation model training, validation and evaluation, respectively, the latter of which were not used during model optimization. These data consisted of the image data as well as of the pixel-wise segmentation ground truth (combined representation in Fig 4).

## Segmentation classes

The segmentation ground truth differentiated two classes: 1.) "diatom", which referred to pixels being part of relevant diatom objects, i.e. valves or frustules which were at least 75% complete and thus had been annotated manually (highlighted in red in Fig 4), and 2.) "background", which referred to anything not being of class "diatom". The latter comprised pixels depicting no object at all, any non-diatom object like e.g. silt or clay particles, dense diatom aggregates as well as diatom girdle bands and smaller, irrelevant diatom fragments which would not be counted or identified in a standard diatom analysis.

## Image data tiling methods

To circumvent scaling down the image data and segmentation ground truth for processing, the training, validation and evaluation image subsections were split into tiles of 512 x 512 pixel, complying with the input size of the segmentation models. Three different tiling approaches were used as follows:

1. <u>Fixed-stride tiling</u>: The image data was covered by a sliding window approach, using a fixed stride of 256 pixel. This places a tile at an overlap of 50% with each adjacent tile in the horizontal as well as in the vertical direction. As a result, each object was depicted at least partially in a minimum of four different tiles (Fig 4). This approach was used on training, validation and evaluation data and was implemented in C# for training and validation data, and in Python for evaluation data.

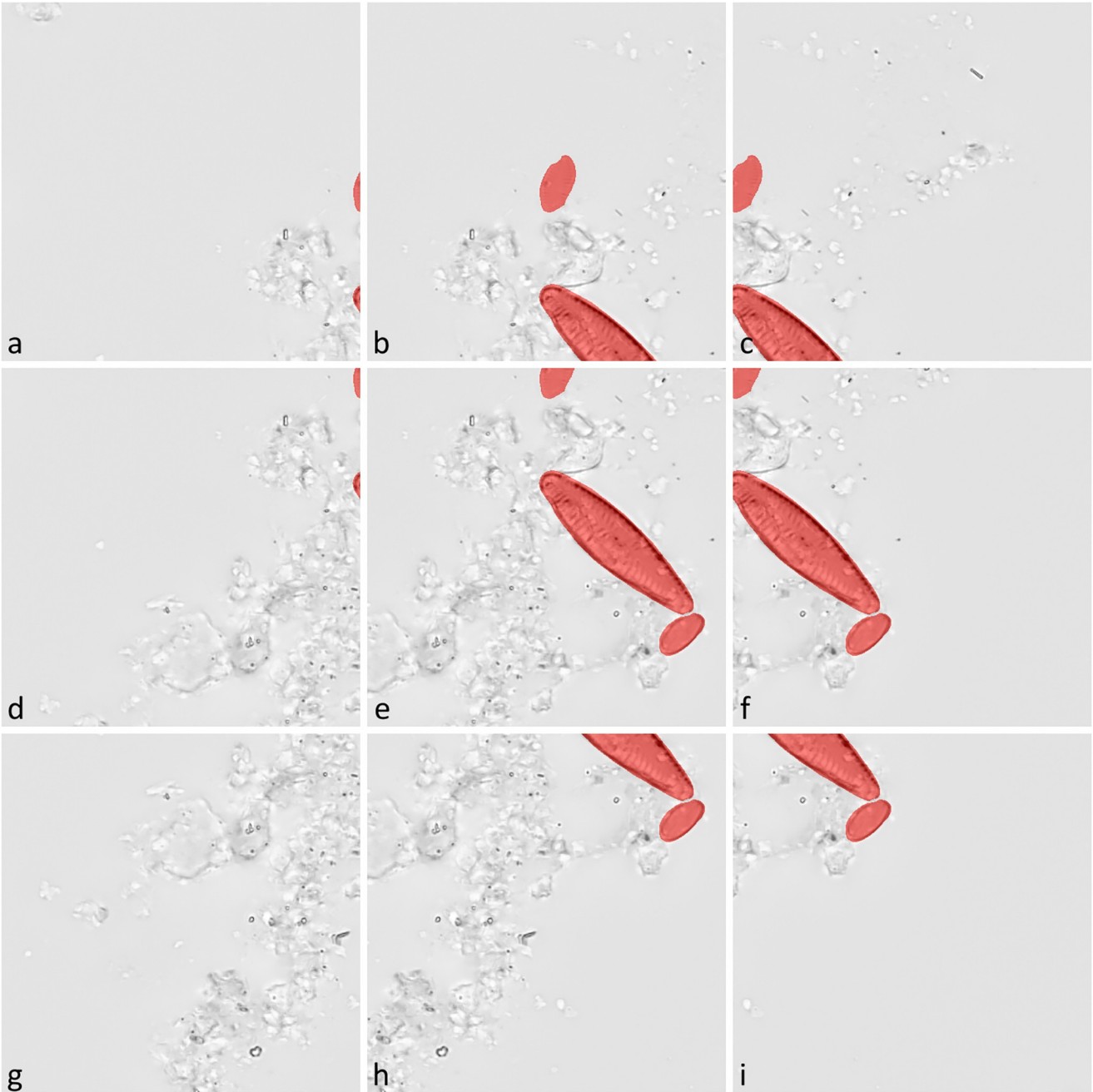

**Fig 4. Example of tiles generated by the fixed-stride tiling method.** Red highlights mark areas which were used to define the segmentation mask (ground truth); objects might be truncated arbitrarily by the tile margins (a-f, h, i). Tiles not containing any diatom object were excluded from the training data (g).

2. <u>Object-based positioning</u>: Tiles were positioned with respect to the boundaries of the man-ually annotated, SHERPA-refined outline of each relevant diatom object. In each tile the object was justified horizontally either to the left, center, right and vertically either to the top, center, bottom, respectively, factoring in a margin of 10 pixel between the object and the tile boundaries. As a result, each object was depicted at least partially in the nine different tiles based on its position (Fig 5, colored highlights), but might also be (partially)

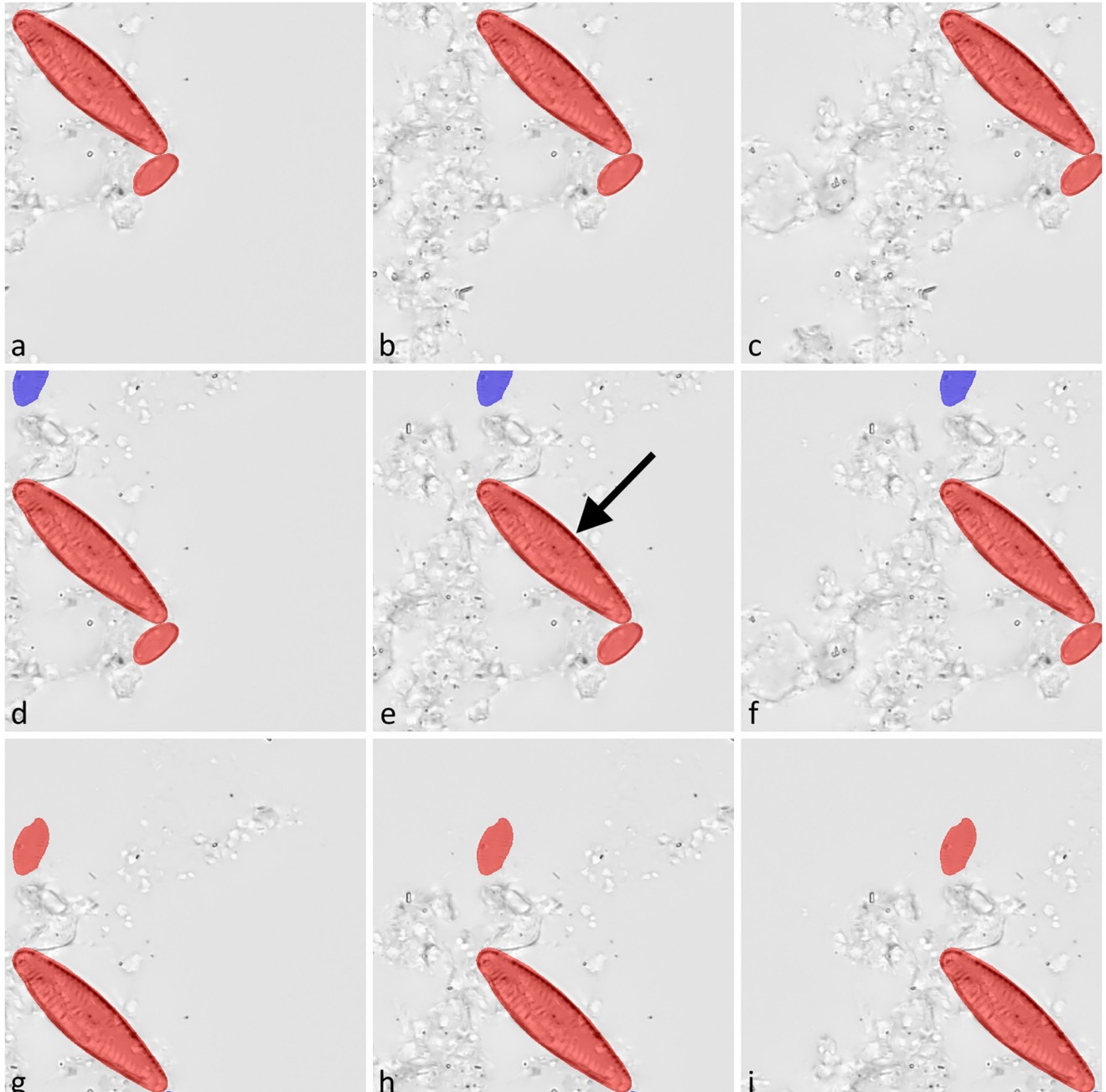

**Fig 5. Training data tiles generated by the two object-based positioning methods for one specific diatom object (marked with an arrow in e).** The two smaller diatom objects were contained coincidentally due to their close proximity. For each annotated object, nine tiles were created (a-i). Object-based positioning alone incorporated all objects in the segmentation mask (red and purple highlights). Contrasting this, object-based positioning + object integrity constraint excluded neighboring objects from a tile's segmentation mask if they were contained by less than 75% within that tile (d-i, purple highlights).

included in tiles based on neighboring objects. This tiling method was only used on training data and was implemented in C#.

3. Object-based positioning + object integrity constraint: Tile positions were the same as with object-based positioning, but neighboring relevant diatom objects, which were

coincidentally contained within a tile's image data because of their spatial proximity, were considered in the segmentation mask only if at least 75% of their area were encompassed by the tile (Fig 5, included objects are highlighted in red, excluded objects in purple color). The area of a diatom object within the tile or the ground truth, respectively, was determined by counting the number of corresponding pixels. As a result, initially relevant diatom objects were deemed irrelevant if they were substantially cropped by a tile's boundaries, which can happen in a substantial number of cases depending on the material density and the size of diatoms. Such cropped objects resemble small valve fragments, which are not supposed to be considered as relevant in the segmentation ground truth. The motivation behind the selected size threshold is to mimic an approach often used in diatom abundance counts e.g. in biomonitoring, where only frustules or valves are considered which are at least about 75% intact [39, 40]. An exception to this rule was made for objects that were larger or wider than the tile and thus probably always would be excluded by the 75% rule. Those were considered if they were covered by at least two of the tile's quarters. This tiling method was used only on training data and was implemented in C#.

## Preparation of training and validation data

For each tile, image data as well as the corresponding segmentation mask were prepared in the form of TIFF files. Subsequently, the segmentation mask of each tile was converted into the COCO data format [43] which described the outline of each contiguous area depicting diatom objects.

Training data were prepared using the fixed-stride as well as both object-based tiling methods in order to conduct a comparative study on the impact of the three approaches on the segmentation performance. Validation data were prepared using only the fixed-stride method to emulate production time inference, where segmentation ground truth would be unknown and hence the object-based tile positioning methods would not be applicable.

About 84% of the tiles generated with the fixed-stride method did not contain any diatom objects at all (such as Fig 4g) due to sparse density of diatoms in the virtual slide images. These tiles were excluded from the training and the validation data to reduce imbalance between the "diatom" and "background" class and to allow for training the Mask R-CNN models using the Detectron2 framework (see below for details), which does not support training and validation data that do not contain any objects. The above-mentioned processing steps were implemented in Python (S2 Dataset).

The data sets resulting from these tiling strategies are described in Table 1. In some of our experiments (see below) the training set size was reduced to 10, 25 and 50% of the diatom objects by horizontally clipping the corresponding training subsection of the virtual slide image. As a consequence of the individual tile positioning approaches, training sets for the three tiling methods comprised the same objects, but the number of tiles was on average roughly three times higher for the object-based approaches.

## Deep learning setup

Model training and evaluation were conducted on a computing server running Ubuntu 18.04.4 LTS, equipped with an NVIDIA® V100 GPU with 32 GB RAM running CUDA v10.2 [44]. Python (v3.6.9) scripts were implemented in the form of Jupyter Notebooks [45], using detectron2 (v0.2.1) [46], PyTorch (v1.6.0) [47], Tensorflow (v2.3.0) [48] and keras-unet (v0.1.2) [49] packages.

**Table 1. Description of data sets.**

| Data set usage | Tiling method | Subset size | Tiles | Diatom objects (approx.) |
|---|---|---|---|---|
| Training | fixed-stride | 100% | 5,869 | 1,750 |
| | | 50% | 2,753 | 870 |
| | | 25% | 1,413 | 470 |
| | | 10% | 481 | 175 |
| | object-based positioning | 100% | 15,732 | 1,750 |
| | | 50% | 7,735 | 870 |
| | | 25% | 3,733 | 470 |
| | | 10% | 1,357 | 175 |
| | object-based positioning + object integrity constraint | 100% | 15,732 | 1,750 |
| | | 50% | 7,735 | 870 |
| | | 25% | 3,733 | 470 |
| | | 10% | 1,357 | 175 |
| Validation | fixed-stride | 100% | 1,652 | 580 |

## Segmentation model architecture

Two basic segmentation model architectures were investigated: 1.) Mask R-CNN, utilizing the Detectron2 framework based on PyTorch, and 2.) U-Net, utilizing the keras-unet package based on Tensorflow / Keras. In preliminary experiments we investigated different backbones and learning rate schedules for Mask R-CNN, as well as different normalization-techniques and layer depths for U-Net, to determine the most suitable model variants. The best performing Mask R-CNN model was"mask_rcnn_X_101_32x8d_FPN_3x"from the Detectron2 model zoo [50], pre-trained on the COCO 2017 data set [43], and an initially untrained "satellite-unet" model with default settings from the keras-unet package (see S2 Dataset for implementation details).

## Model training and validation

The Mask R-CNN model was trained for 5,000 iterations with a batch size of 16 using the model's default loss function and applying flipping and mirror augmentation. The model performance on the validation data was monitored by the segmentation AP metrics of the Detectron2 COCO evaluator. Since validation AP converged after a maximum of about 2,000 training iterations regardless of the experimental setup (see below), the model obtained at the end of the training was used for subsequent evaluation.

The U-Net model was trained for 100 epochs with a batch size of 16 using the Adam optimizer with 1-Dice's coefficient as loss function and applying flipping and mirror augmentation. The loss converged within this 100 epochs, and the model from the epoch obtaining the lowest loss on the validation data was used for subsequent evaluation.

## Experimental setup

We conducted a full factorial experiment to assess the influence of four different factors on segmentation performance: 1.) architecture of the segmentation model, 2.) tiling method, 3.) training data set size and 4.) the threshold applied to the predicted segmentation score to rate an object prediction as valid. This resulted in a total of 72 combinations (Table 2), which were tested on each of the six different virtual slides.

**Table 2. Factors and levels of the full factorial experimental design.**

| Factor | Model architecture | Tiling method | Size of training data set | Threshold prediction score |
|---|---|---|---|---|
| Levels | 1. Mask R-CNN<br>2. U-Net | 1. Fixed-stride<br>2. Object-based<br>3. Object-based positioning + object integrity constraint | 1. 10%<br>2. 25%<br>3. 50%<br>4. 100% | 1. 0.90<br>2. 0.95<br>3. 0.98 |

## Model evaluation

Model segmentation performance was evaluated on data unknown to the model (leave-out-data) containing approximately 580 diatoms objects. The fixed-stride tiling method was applied to the evaluation subsection of each virtual slide image to generate the evaluation data (see S2 Dataset). In contrast to training and validation data, tiles that did not contain any diatom objects were retained. This approach mimicked a production system for virtual slide based diatom segmentation, where the position of potential diatom objects would be unknown a priori. To ensure that each pixel was covered by four different tiles, reflection padding was applied to the tiles exceeding the margins of the evaluation subsection.

Each tile was fed as input image into the segmentation models. For each individual diatom object detected within the input, the Mask R-CNN models produced a segmentation result in form of an outline as well as the corresponding prediction score. These were fused into a single output matrix which contained the prediction score for belonging to the diatom class for each input pixel. This matrix is analogous to the direct output of the U-Net models.

To reverse the tiling process, the prediction score matrices for all input tiles were then assembled into one large matrix of the same size as the corresponding evaluation subsection of the virtual slide image. Each pixel's prediction score was obtained from the maximum prediction score that pixel received in the four overlapping tiles it was contained in. Subsequently, a threshold of 0.90, 0.95 and 0.98 was applied to the segmentation score to generate a segmentation mask.

Each model's segmentation performance was evaluated by comparing the segmentation result to the corresponding ground truth, using pixel-based precision and recall as well as Dice's coefficient (Eqs 1–3). We abstained from using the also common Jaccard index because it is very similar and correlates to the Dice's coefficient.

$$Precision = \frac{TP}{TP + FP} \tag{1}$$

$$Recall = \frac{TP}{TP + FN} \tag{2}$$

$$\begin{aligned} Dice\text{'s coefficient} \quad &= \frac{2 \times Area\ of\ Intersection}{Area\ of\ Union + Area\ of\ Intersection} \\ &= \frac{2 \times TP}{TP + FN + FP + TP} \end{aligned} \tag{3}$$

With TP = number of true positive pixel predictions, FN = number of false negative pixel predictions, FP = number of false positive pixel predictions.

**Table 3. Effects of tiling method, training data set size and prediction threshold on the segmentation performance of the Mask R-CNN models.**

| Factors | Dice's Coeff. | Recall | Precision |
|---|---|---|---|
| **Baseline (tiling: fixed stride; data set size: 10%; prediction threshold: 0.9)** | **= 0.701** *** | **= 0.880** *** | **= 0.591** *** |
| Tiling: object based | + 0.033 *** | - 0.008 | + 0.052 *** |
| Tiling: object-based positioning + object integrity constraint | + 0.071 *** | - 0.024 *** | + 0.120 *** |
| Data set size: 25% | - 0.002 | + 0.022 ** | - 0.015 |
| Data set size: 50% | - 0.003 | + 0.041 *** | - 0.026 |
| Data set size: 100% | - 0.015 | + 0.050 *** | - 0.048 ** |
| Prediction threshold: 0.95 | + 0.020 * | - 0.014 · | + 0.035 * |
| Prediction threshold: 0.98 | + 0.040 *** | - 0.040 *** | + 0.083 *** |

The indicated significance is based on three-factor additive linear model fits. First line (highlighted in bold) indicates the baseline value of the respective metric, lines below that the respective linear model coefficients, i.e., the effects upon the metrics relative to the baseline.

Significance codes:

*** $p < 0.001$,

** $p < 0.01$,

* $p < 0.05$,

· $p < 0.1$

## Results and discussion

A total of ca. 4 gigapixel of image data, depicting roughly 3,000 specimens belonging to about 110 taxa, was used for training, validating and evaluating two types of segmentation models. The data comprised subsections of six virtual slide images, each ca. 55,000 by 1,700–39,000 pixel, depending on the diatom density of the particular sample. For each virtual slide image, these were divided into three subsets by a split of 60:20:20% for generating training, validation and evaluation data, respectively, which were then pooled into the corresponding data sets. The training subsections were then subsampled into four different dataset sizes and processed using three different tiling strategies.

Models were generally able to differentiate between relevant diatoms and irrelevant non-diatom objects / background. The segmentation performance was clearly impacted by the different factor levels (Tables 3 and 4, S1 Table, S1 Fig). Object-based tile positioning, especially if

**Table 4. Effects of tiling method, training data set size and prediction threshold on the segmentation performance of the U-Net models.**

| Factors | Dice's Coeff. | Recall | Precision |
|---|---|---|---|
| **Baseline (fixed stride tiling; data set size: 10%; prediction threshold: 0.9)** | **= 0.485** *** | **= 0.614** *** | **= 0.445** *** |
| Tiling: object based | + 0.110 *** | + 0.043 *** | + 0.130 *** |
| Tiling: object-based positioning + object integrity constraint | + 0.114 *** | + 0.001 | + 0.169 *** |
| Data set size: 25% | + 0.132 *** | + 0.179 *** | + 0.068 ** |
| Data set size: 50% | + 0.126 *** | + 0.199 *** | + 0.057 * |
| Data set size: 100% | + 0.208 *** | + 0.232 *** | + 0.152 *** |
| Prediction threshold: 0.95 | + 0.000 | - 0.000 | + 0.001 |
| Prediction threshold: 0.98 | + 0.000 | - 0.001 | + 0.001 |

The indicated significance is based on three-factor additive linear model fits. First line (highlighted in bold) indicates the baseline value of the respective metric, lines below that the respective linear model coefficients, i.e., the effects upon the metrics relative to the baseline.

Significance codes:

*** $p < 0.001$,

** $p < 0.01$,

* $p < 0.05$

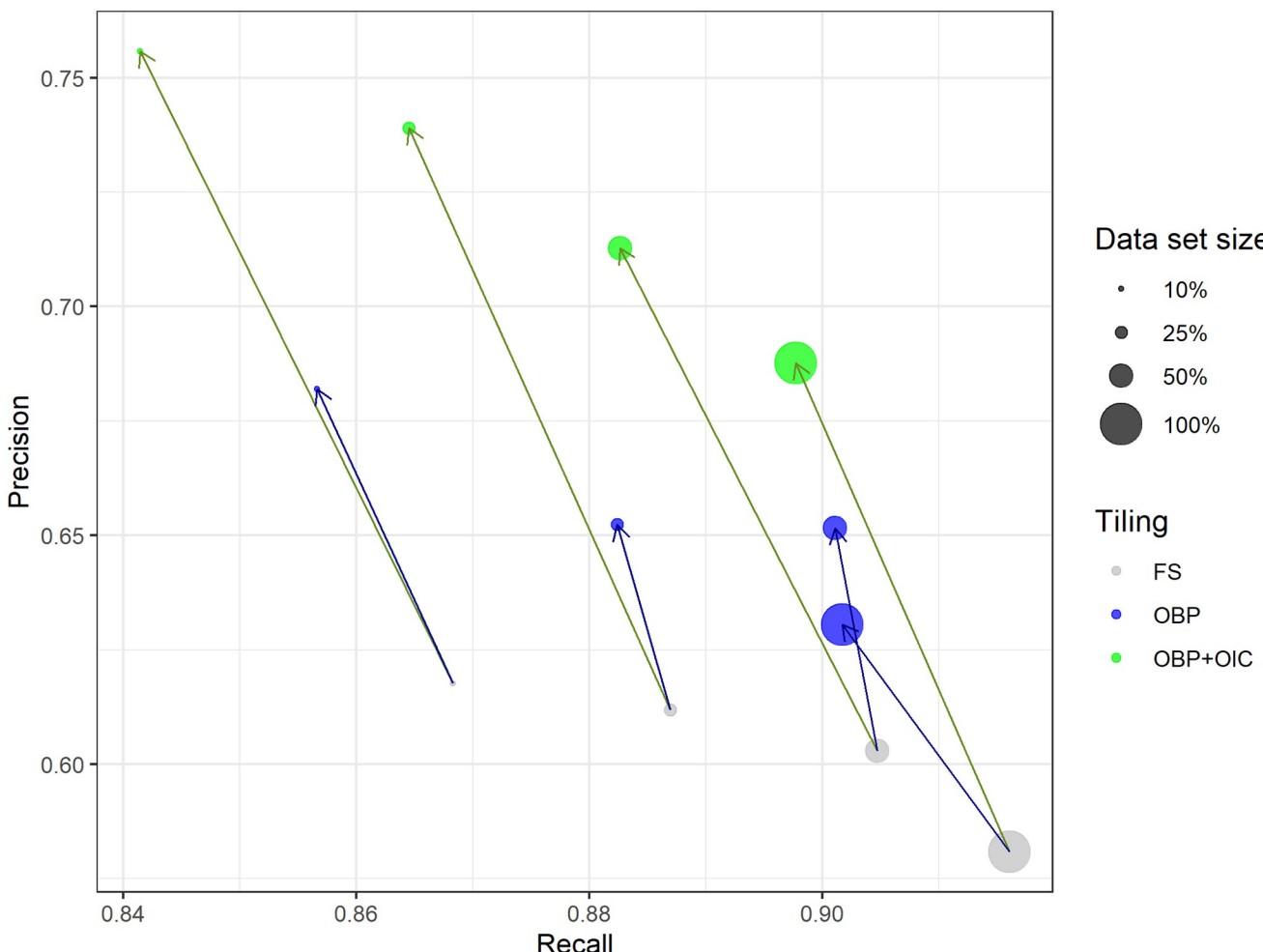

**Fig 6. Effect of training data set size and tiling method on segmentation performance for Mask R-CNN models with prediction threshold 0.95, average over six different virtual slides.** Blue arrows indicate shift from fixed-stride to object-based tile positioning, green arrows indicate shift to object-based positioning + object integrity constraint. FS = fixed-stride, OBP = object-based positioning, OBP+OIC = object-based positioning + object integrity constraint. Please note the different axis scalings, i.e., that substantial improvements in precision with the object-based tiling schemes are usually associated with relatively much smaller decreases in recall.

combined with object integrity constraint, had beneficial effects on the segmentation precision in general (Figs 6 and 7, S2–S4 Figs).

## Mask R-CNN

An ANOVA with additive effects (Table 3) showed significant impact of a) object-based positioning as well as of object-based positioning + object integrity constraint tiling, b) larger training set sizes and c) higher prediction score thresholds, compared to the baseline (10% of total training data, fixed-stride tiling and a prediction threshold of 0.9).

Object-based positioning improved Dice's coefficient from 0.701 to 0.734 and precision from 0.591 to 0.643 ($p < 0.001$ for each case), without impacting the recall of 0.880 significantly. We attribute this to an improved ability of the models to tell apart diatom from non-diatom objects due to training data augmented by shifting the image data, without introducing any geometric distortion like cropping, scaling or warping.

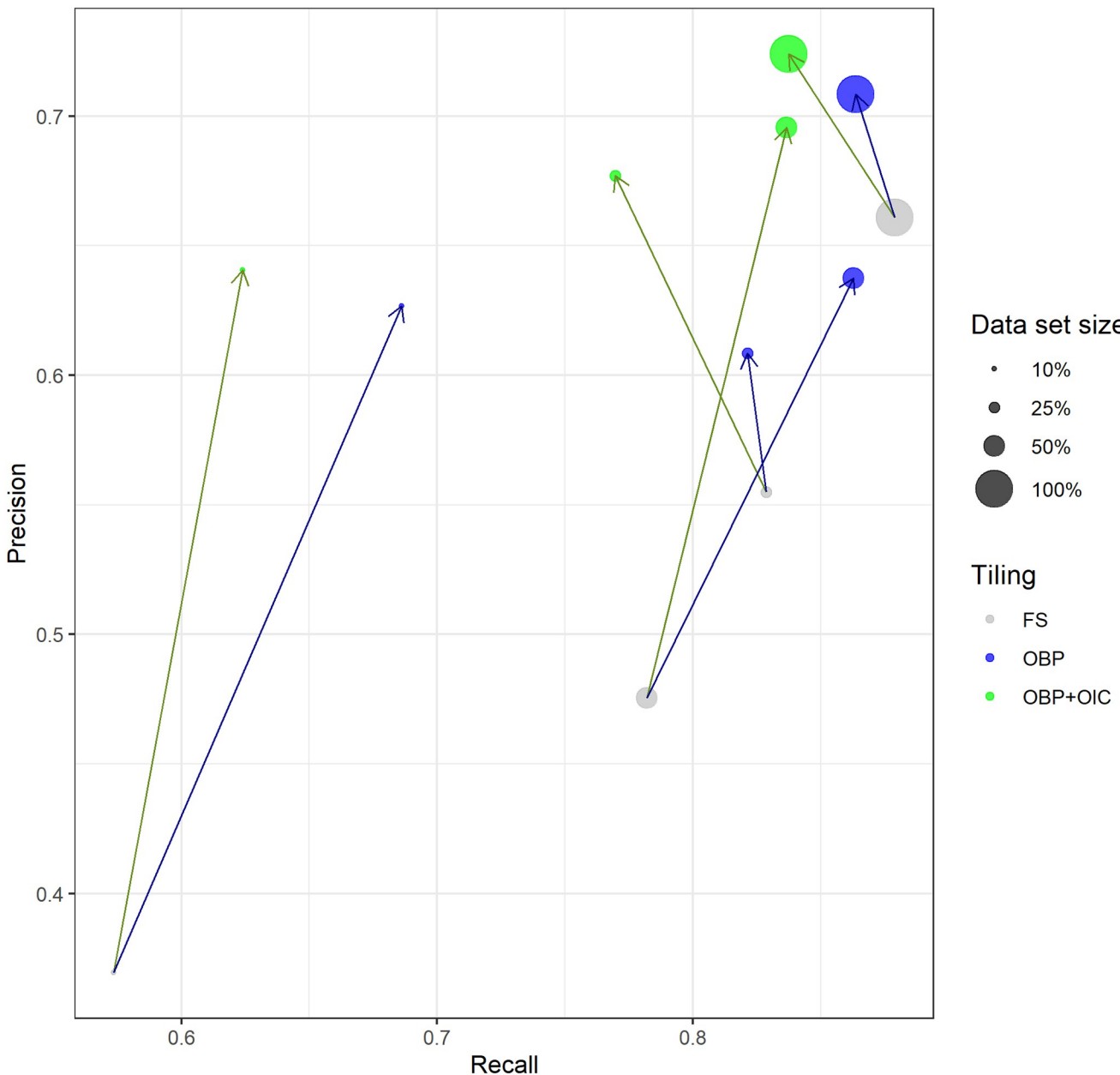

**Fig 7. Effect of training data set size and tiling method on segmentation performance for U-Net models with prediction threshold 0.95, average over six different virtual slides.** Blue arrows indicate shift from fixed-stride to object-based tile positioning, green arrows indicate shift to object-based positioning + object integrity constraint. FS = fixed-stride, OBP = object-based positioning, OBP+OIC = object-based positioning + object integrity constraint.

Object-based positioning + object integrity constraint improved Dice's coefficient and precision even more, from 0.701 to 0.772 and from 0.591 to 0.711, respectively ($p < 0.001$ for each case). We attribute this to the fact that the models were trained with substantially less cropped diatom objects. This may have had a beneficial effect on the discrimination between relevant diatom objects and a) irrelevant small ($< 75\%$ complete) diatom fragments, as well as from b) non-diatom particles which happen to look similar to such smaller diatom fragments. However, this came at the expense of the recall declining slightly from 0.880 to 0.856 ($p < 0.001$). A

possible reason for this effect is that the models might have been forced to "pay less attention" to some valid diatom features, because these were present in the image data of substantially cropped diatom objects and thus had been excluded from the corresponding segmentation masks.

Increasing <u>training data set sizes</u> from 10% to 100% of all data significantly increased the recall from 0.880 to 0.930 ($p < 0.001$), at the expense of decreasing precision from 0.591 to 0.543 ($p < 0.01$). In general, the Mask R-CNN models already performed well when trained with smaller data sets. This could be explained by the fact that they already had been pre-trained on the COCO 2017 data set and thus were responsive to a broad range of general segmentation features prior to our training.

Increasing the <u>prediction score threshold</u> from 0.90 to 0.98 influenced the segmentation performance, but resulted in the commonly observed tradeoff between recall and precision.

The best performing models were those trained on tiles created by object-based positioning + object integrity constraint, with varying performance depending on data set size and prediction threshold. For our typical application, which aims for a high Dice's coefficient but prefers recall over precision, a prediction threshold of 0.95 is a reasonable choice, resulting in an average Dice's coefficient of 0.775, a recall of 0.898 and a precision of 0.688.

## U-Net

For the U-Net-based segmentation models, ANOVA with additive effects showed strong and significant effects of object-based tile positioning, especially when combined with object integrity constraint, as well as for larger training set sizes, but not for higher prediction score thresholds (Table 4). In general, the segmentation performance was inferior compared to the Mask R-CNN models.

<u>Object-based tile positioning</u> improved Dice's coefficient from 0.485 to 0.595, recall from 0.614 to 0.657 and precision from 0.445 to 0.575 ($p < 0.001$ for each case). This is a stronger impact than for the Mask R-CNN models, but at an overall lower level. Again, we attribute this refinement to an improved ability of the model to discriminate diatom from non-diatom objects due to the augmentation of training data introduced by this tiling scheme.

<u>Object-based positioning + object integrity constraint</u> improved the segmentation performance slightly further, but the effect, compared to sole object-based tile positioning, was much weaker than for the Mask R-CNN models. In total, Dice's coefficient and precision increased from 0.485 to 0.599 and from 0.445 to 0.614, respectively ($p < 0.001$ for each case). Again we attribute this to the fact that the model was trained with substantially less cropped objects, und thus was able to better ignore small diatom fragments and non-diatom particles similar to them. In contrast to pure object-based tile positioning, the recall of 0.614 was not affected significantly.

Contrasting with Mask R-CNN models, increasing <u>training data set size</u> substantially enhanced the segmentation performance of the U-Net models. Dice's coefficient improved from 0.485 to 0.693, recall from 0.614 to 0.846, and precision from 0.445 to 0.597 ($p < 0.001$ in each case). We hypothesize that this reflects the random initialization of the U-Net models (in contrast to the pre-trained Mask R-CNN). The former usually requires substantially larger training set sizes for the model to learn appropriate segmentation features compared to pre-trained models trained applying transfer learning.

The <u>prediction score threshold</u> had no significant effect on the segmentation performance because most prediction scores above 0.9 were close to 1.0 for the U-Net models.

The best performing model was the one trained on the complete set of data obtained by object-based positioning + object integrity constraint, which reached a recall of 0.837 and a precision of 0.724, resulting in a Dice's coefficient of 0.775 (S1 Fig).

## Segmentation output

All models showed the ability for discriminating diatom objects from background containing non-diatom objects even on image material where traditional threshold-based segmentation methods failed. We observed (though did not quantify) the latter for the vast majority of diatom objects during data preparation, when reworking object outlines using our SHERPA tool [13], which implements Otsu's thresholding [51], Canny edge detector [52], robust automated threshold detector [RATS, 53] and adaptive thresholding [54]. Models trained on data obtained by object-based tile positioning strongly outperformed those trained with fixed-stride tiling, and segmentation performance improved even more with object integrity constraint. By visual inspection of the segmentation results, we mainly attribute this to a better discrimination of mostly intact diatom objects from small diatom fragments or non-diatom objects resembling them (Fig 8b–8d). Both, discriminating diatom objects from non-diatom ones as well as from small diatom fragments, are essential prerequisite for a high-throughput automated identification and counting.

We observed that individual diatom objects in aggregates were mostly recognized as long as they were clearly visually separated from each other. Nevertheless, this separation was hampered by our application of focus stacking, which drastically increased focal depth at the expense of no longer being able to sort out aggregates by isolating objects on different focal planes. Also, our approach of integrating the results from overlapping tiles precluded the separation of overlapping or touching cells, even if they originally had been disaggregated by instance segmentation executed by the Mask R-CNN models. This was a deliberate decision to facilitate comparison between the two segmentation model types we investigated, where U-Net only performs semantic segmentation, which per se cannot separate overlapping individual objects.

In the presented experiments and settings, recall was higher than precision, which is preferable for investigations that favor to not miss diatom objects over investing additional efforts for sorting out false positives by manual or automated downstream processing. Visual inspection of segmentation results (Figs 9–11, S3 Dataset) showed that in general only few diatom objects were overlooked. Whilst the Mask R-CNN models had a tendency of overlooking or untruly predicting complete diatom objects (Figs 9 and 10), U-Net models often overlooked only subsections of true diatom objects (Fig 11), which might be favorable over missing them completely. For Mask R-CNN models, the balance between precision and recall had to be tuned by selecting an appropriate segmentation score threshold, resulting in either more overlooked diatom objects or more false positive predictions (Figs 9 and 10). For both model architectures, one common reason for false positives was the erroneous detection of diatom fragments which were of substantial but still too small size ($< 75\%$ complete) to be considered during a standard aquatic ecology or biomonitoring analysis. This problem was alleviated by object integrity constraint. Another source of false positives were diatom objects which were contained within very heterogeneous areas of the background, which actually comprised sediment particles, small diatom fragments or dense aggregates of diatom material. The latter had been excluded from segmentation ground truth if it was not possible to accurately tell apart individual objects manually. However, in some cases the segmentation models might have surpassed the human annotators here, and some of the alleged false positives might actually have been true positives. Similarly, during initial test runs, the segmentation models detected lightly

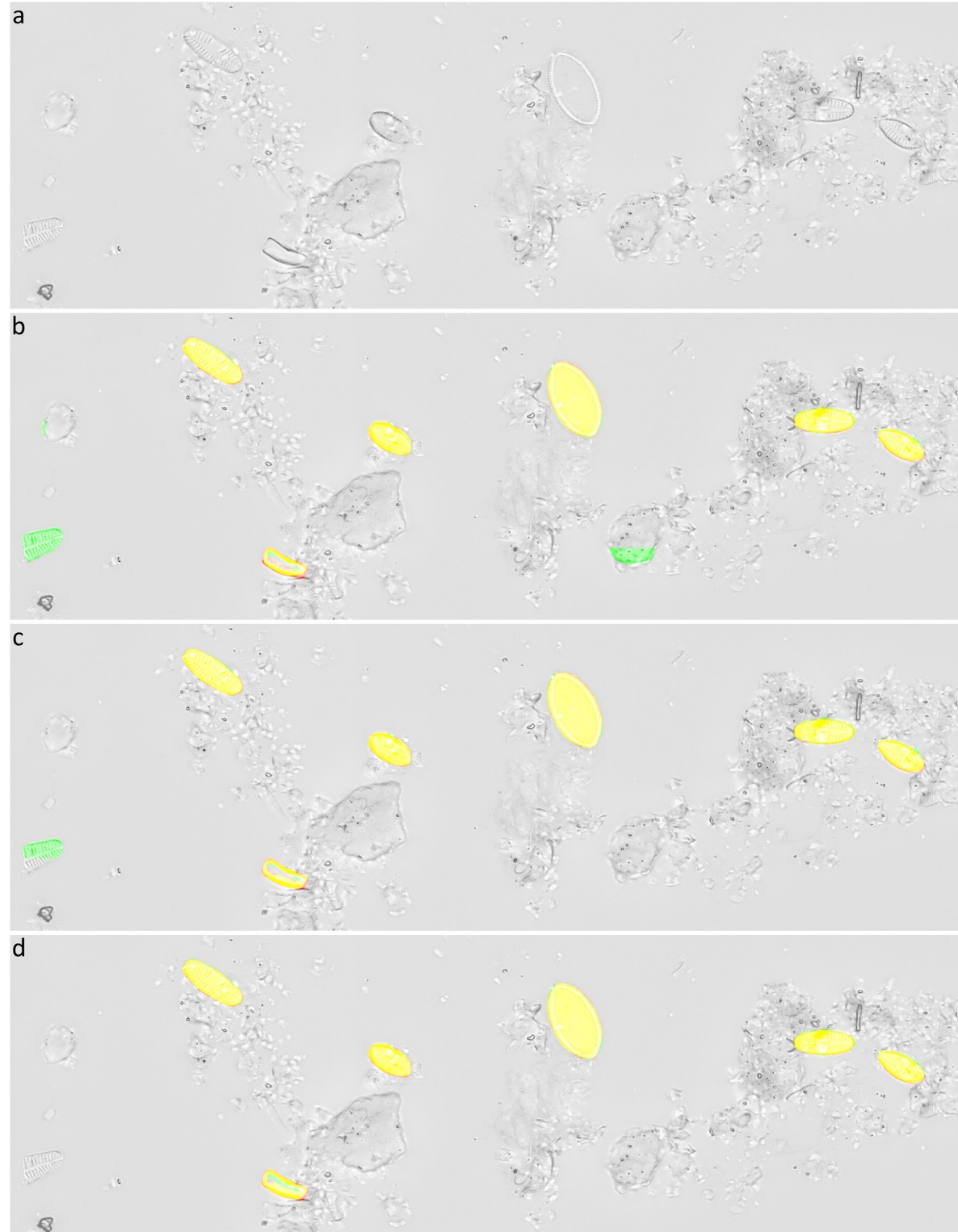

**Fig 8. Typical segmentation results of Mask R-CNN models trained on 100% training data, with a prediction score threshold of 0.95.**
a) Input data, b) segmentation result for fixed-stride training data, c) segmentation result for object-based training data, d) segmentation result for training data generated by object-based positioning + object integrity constraint. Yellow = true positive, red = false negative, green = false positive.

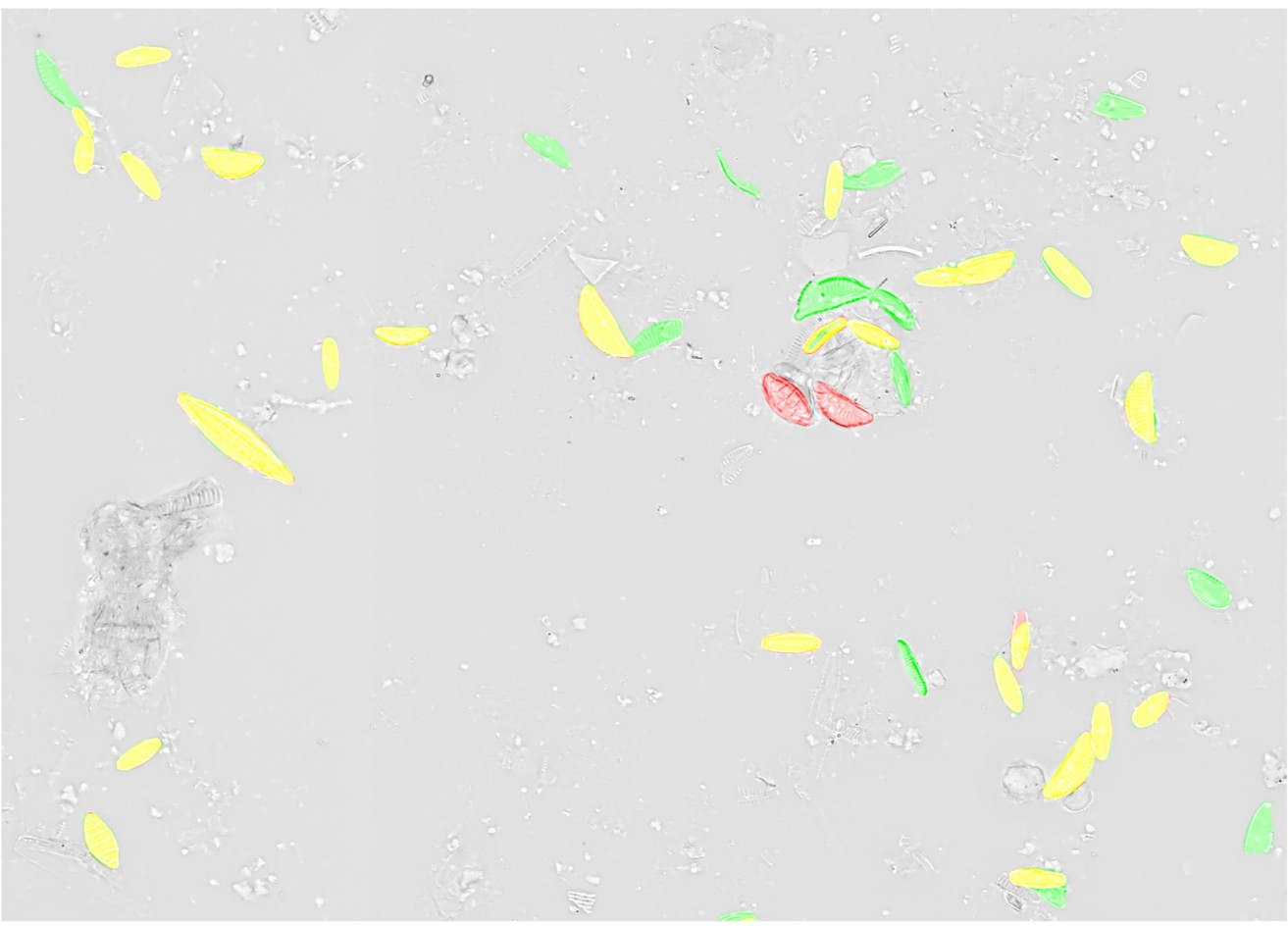

**Fig 9. Comparison of predicted segmentation vs. ground truth for a Mask R-CNN model (object-based tile positioning + object integrity constraint, 100% training data) applied with a prediction score threshold of 0.9 on previously unseen test data.** Yellow = true positive, red = false negative, green = false positive.

silicified diatom objects which initially had been missed by human annotators due to their very poor contrast (Fig 12). After this observation, the segmentation ground truth was extended by these additional low-contrast objects and scrutinized visually to improve its quality prior to the final experiments reported above. These observations, combined with the above presented results, altogether suggest that in terms of recall, the CNN-based segmentation models can perform comparably or in some difficult instances even better than a human annotator as far as detecting diatom objects is concerned.

## Comparison Mask R-CNN vs. U-Net

Both model architectures succeeded reasonably well in discriminating relevant, mostly intact diatom objects from the cluttered background. Mask R-CNN models outperformed U-Net models especially for smaller training set sizes, allowing for comparable precision and even better recall with only 10% of training data. Nevertheless, for the Mask R-CNN models only a tradeoff was possible between recall and precision, where a higher prediction score threshold or a lower data set size improved precision at the expense of recall, and vice versa. Contrasting these findings, U-Net models were insensitive to the prediction score thresholds investigated

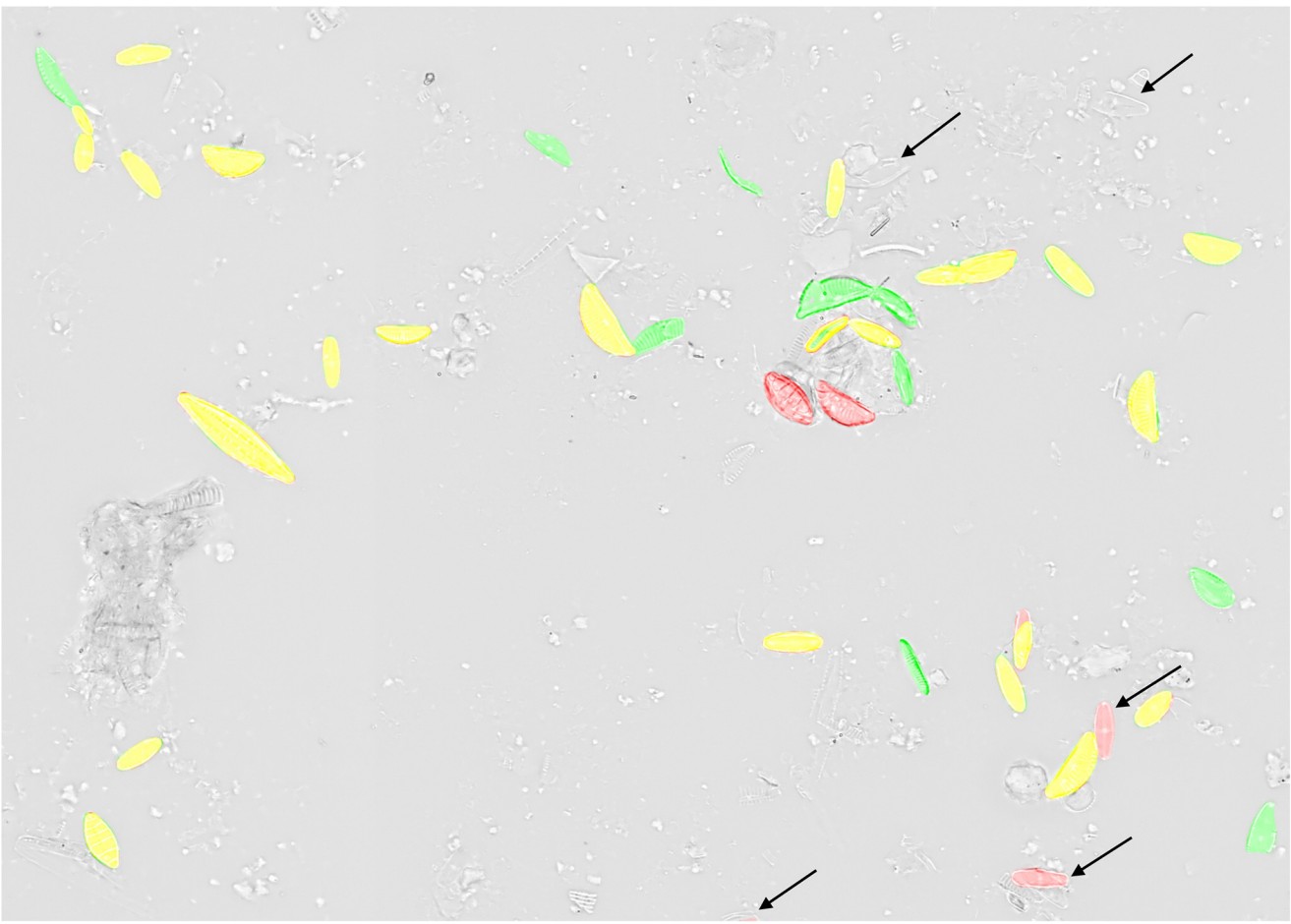

**Fig 10. Comparison of predicted segmentation vs. ground truth for a Mask R-CNN model (object-based tile positioning + object integrity constraint, 100% training data) applied with a prediction score threshold of 0.98 on previously unseen test data.** Black arrows indicate substantial changes in comparison to Fig 9 (prediction score threshold of 0.9). Yellow = true positive, red = false negative, green = false positive.

in this work, but improved in recall as well as in precision with larger training set sizes. For the investigated application, it is thus possible that with even more training data, U-Net models might be able to outperform Mask R-CNN models by allowing higher recall / precision values before causing a tradeoff between them, but it has to be noted that preparing segmentation training data even by our efficient workflow is highly time consuming and thus usually considered a limiting factor. Mirrored padding seems to have reduced the influence of segmentation faults occurring at the margins of image data, especially for the U-Net models. Altogether, when training data are tedious to obtain and represent a major bottleneck, Mask R-CNN models might be preferable.

## Tiling strategy

Object-based tile positioning, especially when combined with object integrity constraint, had a substantial positive effect upon segmentation results with both network architectures tested. As noted above, we attribute this improvement to two causes: 1.) Both of the object-based tiling approaches resulted in roughly 3x more training data tiles then fixed-stride tiling, augmenting the image data by shifting without introducing any geometric distortion like cropping, scaling

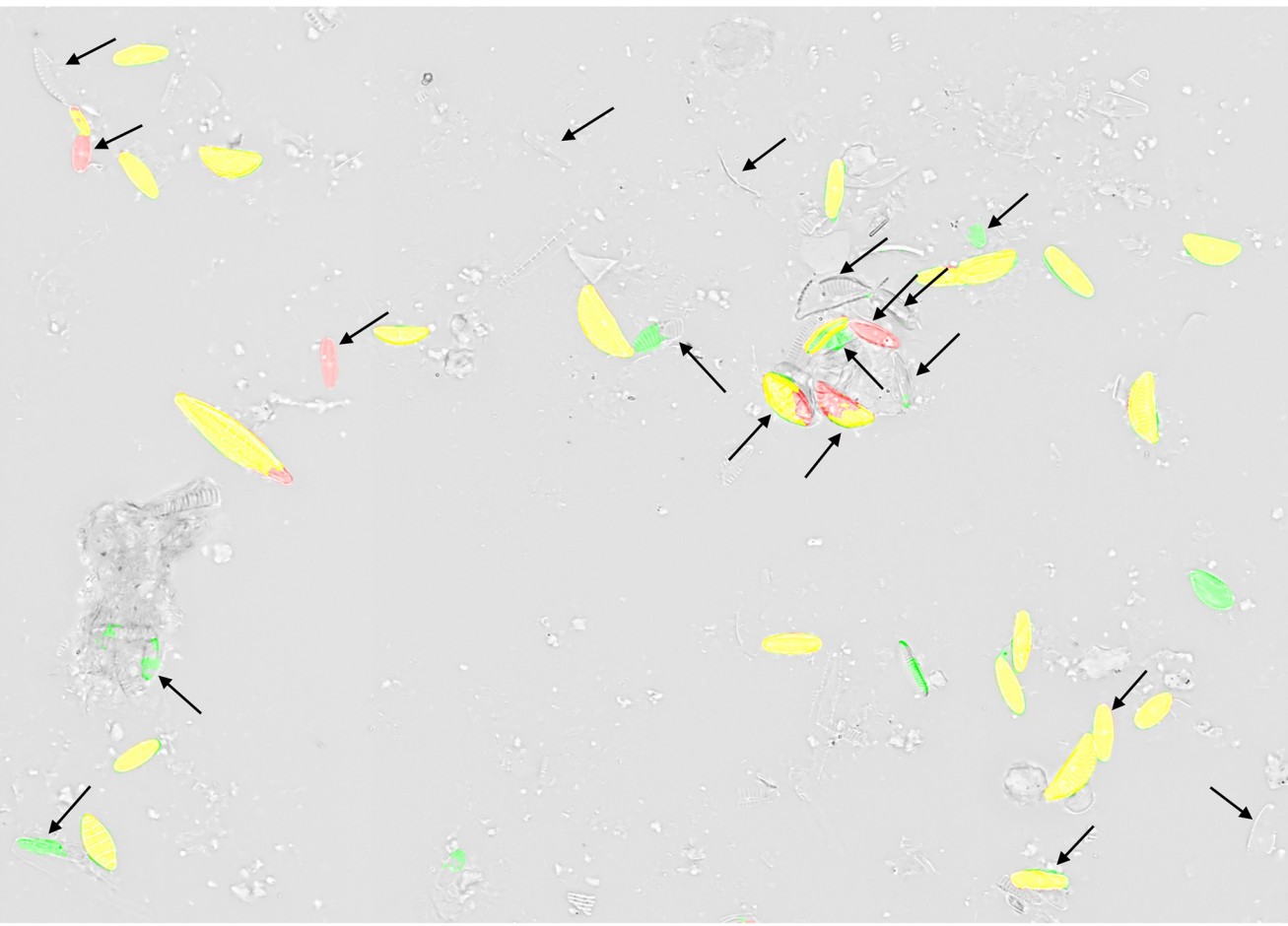

**Fig 11. Comparison of predicted segmentation vs. ground truth for a U-Net model (object-based tile positioning + object integrity constraint, 100% training data, prediction score threshold 0.98) on previously unseen test data.** Black arrows indicate substantial changes in comparison to Fig 10 (same prediction score but Mask R-CNN model). Yellow = true positive, red = false negative, green = false positive.

or warping. 2.) The combination of object-based positioning with object integrity constraint took this effect even further by eliminating diatom objects from the training data segmentation masks that were cropped by a tile's boundary and thus accidentally resembled small diatom fragments. This improved the model's ability to discriminate relevant, mostly intact diatom objects from irrelevant objects such as small diatom fragments and similar looking non-diatom particles. As a result, we observed a substantial and significant increase of segmentation precision, but only little effect on recall.

## Conclusions

Scanning large areas of microscope slides generating gigapixel-sized virtual slide images, and the automated segmentation thereof, are essential prerequisites for high-throughput automated diatom identification and enumeration. Imaging can now be performed utilizing commercial slide scanning microscopes. Deep learning-based segmentation of large scans of permanent diatom slides is feasible by utilizing image tiling to break the large virtual slide images down into smaller subsections fitting the model's input size. The segmentation results are then stitched into an output corresponding to the original scan size. Contrasting

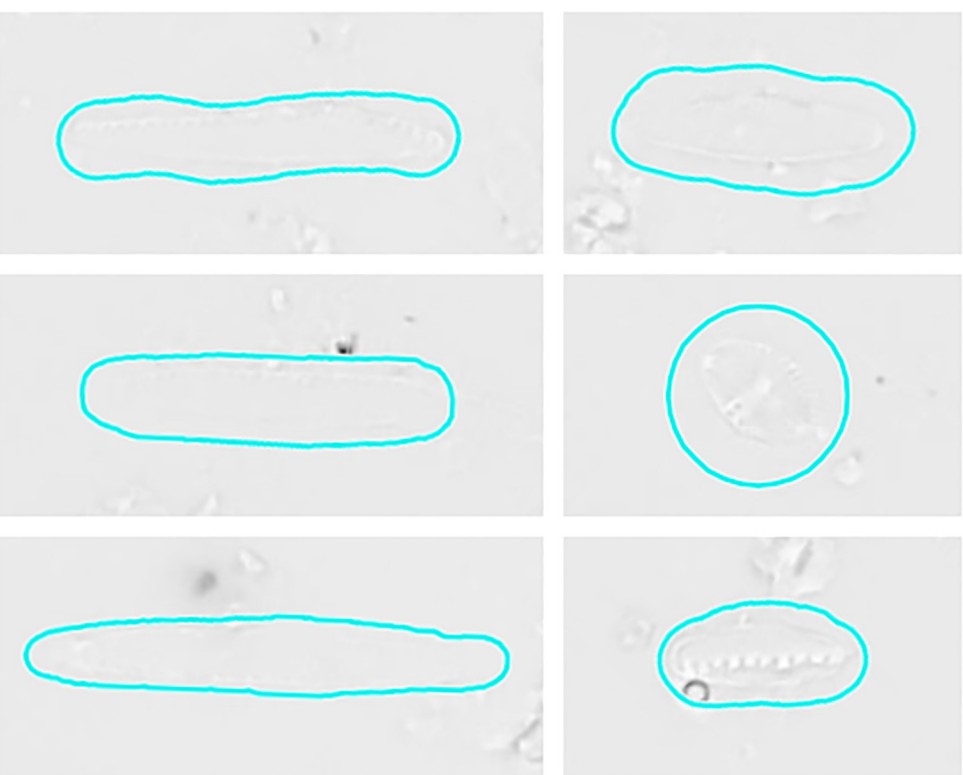

**Fig 12. Examples of relevant diatom objects which had been overlooked by human annotators due to their poor contrast, but were found by our segmentation models during preliminary tests.** For illustration purposes objects are marked very roughly by cyan outline, which is not their actual annotation ground truth.

approaches which scale down image data to the model input size, our approach processes the image data at its original resolution, allowing for highly detailed object masks. These are essential e.g. for shape-based morphometric analyses and might be beneficial for subsequent classification. In the past, data containing a lot of debris or small diatom fragments mostly could not be segmented successfully using methods like thresholding [13]. However, with deep learning-based segmentation models such as Mask R-CNN or U-Net, it has now become possible to pixel-accurately discriminate relevant (i.e. mostly complete) diatom valves / frustules from non-diatom particles or irrelevant small diatom fragments. Thereby, the tiling process might hamper the segmentation performance by cropping originally relevant diatom objects and thus accidently turning them into training data which visually appear similar to irrelevant small fragments. This is at least partially mitigated by object-based tile positioning approaches, which also might generate training benefit in form of a simple data shifting augmentation. Training data created with the object-based tile positioning + object integrity constraint introduced by this work reduces the amount of randomly cropped objects in the training data even more, and this seems to improve the model´s ability to ignore small irrelevant fragments without unlearning relevant diatom objects. Even though we developed this tiling strategy specifically for preparations of oxidized diatom frustules, it is likely to improve segmentation also in other areas, especially for objects which are small enough to fit into a single tile, but large enough to often be truncated by a tile's border (potentially including large-scale live cell imaging of diatoms or other microscopic organisms). Pre-trained Mask R-CNN models, which are available e.g. from the Detectron2 model zoo, allow for high segmentation performance by

transfer learning with only a fraction of training data and roughly 10x faster training compared to the full data set U-Net models, hence they are a recommendable solution. One has to keep in mind that the taxonomic composition of the training data might limit the applicability of any segmentation model to target taxa with highly different morphologies. For this reason, it is presently unclear if completely generic diatom segmentation models will be available in the future; otherwise, care will need to be taken to use training data ideally close to the taxonomic composition of the target communities. Altogether, and keeping in mind such caveats, slide scanning microscopy combined with segmentation methods like those presented here, and with convolutional classifier networks for taxonomic identification as tested on diatoms in numerous instances [31–33, 55], can in the near future potentially provide the possibility for large-scale imaging based diatom community analysis.

## Supporting information

**S1 Dataset. Virtual slide images of diatom preparations from river Menne.** https://doi.org/10.5281/zenodo.7078938.
(DOCX)

**S2 Dataset. Python scripts / Jupyter Notebooks and data for training segmentation models on virtual slide images of diatom preparations from river Menne.** https://doi.org/10.5281/zenodo.7079005.
(DOCX)

**S3 Dataset. Mask R-CNN and U-Net segmentation models and experimental results on segmenting virtual slide images of diatom preparations from river Menne.** https://doi.org/10.5281/zenodo.7079072.
(DOCX)

**S4 Dataset. Segmentation performance evaluation results and R scripts for generating Figs 5 and 6 as well as Tables 3 and 4.** https://doi.org/10.5281/zenodo.7107456.
(DOCX)

**S1 Fig. Boxplots of segmentation performance scores.** Investigated factors are model architecture, tiling method (FS = fixed-stride, OBP = object-based positioning, OBP+OIC = object-based positioning + object integrity constraint), training data set size (coded in the y-axis labels) and prediction threshold (coded in colors). Crosses depict mean, vertical lines within boxes median values.
(PDF)

**S2 Fig. Effect of training data set size and tiling method on segmentation performance for Mask R-CNN models with prediction threshold 0.95 on six different virtual slides.** Blue arrows indicate shift from fixed-stride to object-based tile positioning, green arrows indicate shift to object-based positioning + object integrity constraint. FS = fixed-stride, OBP = object-based positioning, OBP+OIC = object-based positioning + object integrity constraint. Please note the different axis scalings, i.e., that substantial improvements in precision with the object-based tiling schemes are usually associated with relatively much smaller decreases in recall.
(TIFF)

**S3 Fig. Effect of training data set size, tiling method and prediction threshold on segmentation performance for Mask R-CNN models with prediction threshold 0.95, average over six different virtual slides.** Blue arrows indicate shift from fixed-stride to object-based tile positioning, green arrows indicate shift to object-based positioning + object integrity

constraint. FS = fixed-stride, OBP = object-based positioning, OBP+OIC = object-based positioning + object integrity constraint; Pred. threshold = prediction threshold. Please note the different axis scalings, i.e., that substantial improvements in precision with the object-based tiling schemes are usually associated with relatively much smaller decreases in recall.
(TIFF)

**S4 Fig. Effect of training data set size and tiling method on segmentation performance for U-Net models with prediction threshold 0.95 on six different virtual slides.** Results for prediction thresholds 0.90 and 0.98 are nearly identical. Blue arrows indicate shift from fixed-stride to object-based tile positioning, green arrows indicate shift to object-based positioning + object integrity constraint. FS = fixed-stride, OBP = object-based positioning, OBP+OIC = object-based positioning + object integrity constraint. Please note the different axis scalings, i.e., that substantial improvements in precision with the object-based tiling schemes are usually associated with relatively much smaller decreases in recall.
(TIFF)

**S1 Table. Segmentation performance (means over the six virtual slides).** PT = Prediction Threshold, Dice = Dice's coefficient, FS = fixed-stride, OBP = object-based positioning, OBP+OIC = object-based positioning + object integrity constraint.
(PDF)

## Acknowledgments

We wish to thank Marzena Spyra for her support in collecting and processing the samples and Martin Zurowietz for helping with any questions related to BIIGLE. The Open Access Publication Fund of the University of Duisburg-Essen supported publication of this manuscript.

## Author Contributions

**Conceptualization:** Michael Kloster, Daniel Langenkämper, Tim W. Nattkemper, Bánk Beszteri.

**Data curation:** Michael Kloster, Andrea M. Burfeid-Castellanos.

**Funding acquisition:** Tim W. Nattkemper, Bánk Beszteri.

**Investigation:** Michael Kloster.

**Methodology:** Michael Kloster.

**Project administration:** Bánk Beszteri.

**Software:** Michael Kloster.

**Supervision:** Tim W. Nattkemper, Bánk Beszteri.

**Validation:** Michael Kloster.

**Writing – original draft:** Michael Kloster, Andrea M. Burfeid-Castellanos, Daniel Langenkämper, Tim W. Nattkemper, Bánk Beszteri.

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
