## [Decision Letter · Decision Letter 0]

3 Aug 2022

PONE-D-22-19514Improving deep learning-based segmentation of diatoms in gigapixel-sized virtual slides by object-based tile positioning and object integrity constraintPLOS ONE

Dear Dr. Kloster,

Thank you for submitting your manuscript to PLOS ONE. After careful consideration, we feel that it has merit but does not fully meet PLOS ONE’s publication criteria as it currently stands. Therefore, we invite you to submit a revised version of the manuscript that addresses the points raised during the review process.

ACADEMIC EDITOR: Please revise your manuscript based on the reviewers comments and suggestions. 

We look forward to receiving your revised manuscript.

Kind regards,

Kathiravan Srinivasan

Academic Editor

PLOS ONE

Journal Requirements:

3. We noted in your submission details that a portion of your manuscript may have been presented or published elsewhere. [DETAILS AS NEEDED] Please clarify whether this [conference proceeding or publication] was peer-reviewed and formally published. If this work was previously peer-reviewed and published, in the cover letter please provide the reason that this work does not constitute dual publication and should be included in the current manuscript.

Reviewers' comments:

Reviewer's Responses to Questions

**Comments to the Author**

1. Is the manuscript technically sound, and do the data support the conclusions?

Reviewer #1: Yes

Reviewer #2: Yes

Reviewer #3: Yes

Reviewer #4: Yes

2. Has the statistical analysis been performed appropriately and rigorously? 

Reviewer #1: Yes

Reviewer #2: Yes

Reviewer #3: Yes

Reviewer #4: No

3. Have the authors made all data underlying the findings in their manuscript fully available?

Reviewer #1: No

Reviewer #2: Yes

Reviewer #3: Yes

Reviewer #4: Yes

4. Is the manuscript presented in an intelligible fashion and written in standard English?

Reviewer #1: Yes

Reviewer #2: Yes

Reviewer #3: Yes

Reviewer #4: Yes

5. Review Comments to the Author

Reviewer #1: The manuscript discusses some DL-mediated advancements in object segmentation in micrographs through the application of an object-based tiling approach and an object integrity constraint. However, there are some queries I have in this respect:

1. The authors mentioned that the segmentation ground truth only classified a structure as a 'diatom' if at least 75% of it was within a tile. What algorithm was used to compute the surface area of the various diatoms? Was this done computationally via some in-built algorithm in the image acquisition software, or did the author/(s) develop their own algorithm and/or code to do this? If neither (e.g. the authors did this classification manually), how did they ensure consistency between the various diatoms in the sample, when applying this criterion?

2. As the specimen used here was a diatom strew mounted by the author/(s) themselves, how did the currently proposed object-based segmentation approaches perform with regards to diatoms which were stacked/overlaid above other diatoms? The authors mentioned in the manuscript that "very dense aggregates of diatom material, where it was not possible to tell apart individual cells, were not considered" (quote from their manuscript). However, if the aggregates were not as dense (i.e. it was possible to discriminate between the individual cells), but still consisted of different diatom types [e.g. pennate diatoms overlaying rounded diatoms (such as Coscinodiscus sp.) or diatom fragments], how did the proposed object-based segmentation approach perform for such cases? Was it able to pick out the individual diatoms within the pile, or did it treat the overlaid diatoms as a single agglomerated entity?

3. The authors described their developed object-based tiling methods coupled with an object integrity constraint. However, will this influence the stride of the tiles as well, especially if the diatoms are irregularly spaced apart (as in this strew) and how does the algorithm mitigate this effect, since some tiles may contain (for instance) half of a diatom and the next window may contain the other half of the diatom?

4. The authors mentioned about the splitting of the image subsections into 512*512 tiles in the manuscript, but did not mention how this was performed (e.g. using MATLAB, Python, etc). Please include this information in the Methods section as well.

5. As this is a study focused on deep learning, please include information such as the training time (per iteration or epoch) required for training the Mask-RCNN & U-Net models, as well as the loss function plots for each of these frameworks during the training. The former would be useful for gauging the relative computational complexity of the frameworks used, while the latter may be used to identify if the model training proceeded optimally.

6. Please include the registered mark (®) after the word NVIDIA (when referring to the V100 GPU) on page 12.

On a separate note, it would also be good if the authors could include a section on the future potential applications of their current algorithm. Some aspects which the authors might consider to include in this section are as follows:

7. Can this algorithm be used to characterize living diatoms, rather than just the siliceous tests? If unsure, the authors may want to include this point as a statement for future expansion of the current work in optical microscopy.

8. Can the algorithm be developed further to phylogenetically classify the diatoms based on their frustule or poroid composition (down to their individual species)?

Reviewer #2: From a theoretical point of view this new automatic technic of diatom analysis seems promising. "Deep learning" in principle is a milestone in the field of object detection and comparison. But some questions have to be answered:

- The author investigated diatoms from the Menne river comprising more than 110 different species. Please provide a brief introduction about those species. Are there any differences in recall and precision among the species.

- There is little information on the diversity of diatoms analyzed. A model that has been trained from images with similarly shaped diatoms will have more difficulty in detecting other diatom shapes.

- How to ensure the precision of human annotator?

- I suggest that these minor corrections should be done before the publication

Reviewer #3: This paper described an implementation of deploying deep learning-based instance segmentation algorithm, Mask-RCNN, in the area of microorganism detection. It demonstrates that, being well trained on the dataset, Mask-RCNN/U-net have a good performance on recognizing a variety of components and thus can facilitate the analysis of microorganism through image processing. Such applications in biology are not rare (such as Ruiz-Santaquiteria, J., Bueno, G., Deniz, O., Vallez, N., & Cristobal, G. (2020). Semantic versus instance segmentation in microscopic algae detection. Engineering Applications of Artificial Intelligence, 87, 103271).

The authors recommend 3 data preparation/augmentation methods for algorithm training purpose, i.e., 1) fixed-size tiling, 2) object-based positioning, and 3) object-based positioning + object integrity constraint. It was founded that the last method had the best performance. I think the main novelty that the authors intend to present in this paper is the step-by-step improvement among the 3 data augmentation methods. However, to my point of view, the development of these 3 methods is the procedure of eliminating the mistakes made during the process and the improvement in the model performance is no surprise. For example, when the raw image is cut into sub-images, the cutting off objects (below 75%) at the cutting boundary would be wrongly labelled. Expect that the author can prove that this is a routine procedure by the peers, otherwise this error should not have happened during the data preparation and getting rid of such wrongly labelled data should be a normal job (as introduced by the authors in the following 2 methods).

Therefore, my opinion is that the novelty in the paper is not strong enough to convince me to accept it.

Reviewer #4: The research work was lacking originality. The techniques used are well-known, and the contribution made by the use of object integrity constraints is minor and not novel. It's unclear how the object integrity constraint is calculated without first segmenting the object, which appears to be a redundant process.

The database employed is quite small.

6. PLOS authors have the option to publish the peer review history of their article (what does this mean?). If published, this will include your full peer review and any attached files.

Reviewer #1: No

Reviewer #2: **Yes: **Jian Zhao

Reviewer #3: No

Reviewer #4: No

---

## [Author Response · Author response to Decision Letter 0]

23 Sep 2022

Reviewers' comments:

Reviewer's Responses to Questions

Comments to the Author

1. Is the manuscript technically sound, and do the data support the conclusions?

Reviewer #1: Yes

Reviewer #2: Yes

Reviewer #3: Yes

Reviewer #4: Yes

2. Has the statistical analysis been performed appropriately and rigorously? 

Reviewer #1: Yes

Reviewer #2: Yes

Reviewer #3: Yes

Reviewer #4: No

Reviewer 4 did not provide any details in their review about why they thought so (apart from the comment “The database employed is quite small”, which we respond to below), which makes it difficult to address their concerns. Nevertheless, our dataset is considerably larger compared to the publication referred to by Reviewer 3 (Ruiz-Santaquiteria, J., Bueno, G., Deniz, O., Vallez, N., & Cristobal, G. (2020). Semantic versus instance segmentation in microscopic algae detection. Engineering Applications of Artificial Intelligence, 87, 103271), this topic also is addressed below.

3. Have the authors made all data underlying the findings in their manuscript fully available?

The PLOS Data policy <http://www.plosone.org/static/policies.action#sharing> requires authors to make all data underlying the findings described in their manuscript fully available without restriction, with rare exception (please refer to the Data Availability Statement in the manuscript PDF file). The data should be provided as part of the manuscript or its supporting information, or deposited to a public repository. For example, in addition to summary statistics, the data points behind means, medians and variance measures should be available. If there are restrictions on publicly sharing data—e.g. participant privacy or use of data from a third party—those must be specified.

Reviewer #1: No

Reviewer #2: Yes

Reviewer #3: Yes

Reviewer #4: Yes

The data were not available in a public repository at the time of the submission; now they are, as specified in the supplemental section.

4. Is the manuscript presented in an intelligible fashion and written in standard English?

Reviewer #1: Yes

Reviewer #2: Yes

Reviewer #3: Yes

Reviewer #4: Yes

5. Review Comments to the Author

Reviewer #1: The manuscript discusses some DL-mediated advancements in object segmentation in micrographs through the application of an object-based tiling approach and an object integrity constraint. However, there are some queries I have in this respect:

1. The authors mentioned that the segmentation ground truth only classified a structure as a 'diatom' if at least 75% of it was within a tile. What algorithm was used to compute the surface area of the various diatoms? Was this done computationally via some in-built algorithm in the image acquisition software, or did the author/(s) develop their own algorithm and/or code to do this? If neither (e.g. the authors did this classification manually), how did they ensure consistency between the various diatoms in the sample, when applying this criterion?

This remark might be caused by a misunderstanding. The 75% criterion mentioned was applied to the manually annotated ground truth outlines, which are (at training time) known precisely, this is stated in lines 184f, 192, 206 and 228ff. Determining the area of a known ROI in an image simply means counting the number of pixels, so there is no complicated computation involved. The percentage to which the object is contained within a tile is calculated simply by comparing that to the number of object pixels within the tile’s boundaries.

2. As the specimen used here was a diatom strew mounted by the author/(s) themselves, how did the currently proposed object-based segmentation approaches perform with regards to diatoms which were stacked/overlaid above other diatoms? The authors mentioned in the manuscript that "very dense aggregates of diatom material, where it was not possible to tell apart individual cells, were not considered" (quote from their manuscript). However, if the aggregates were not as dense (i.e. it was possible to discriminate between the individual cells), but still consisted of different diatom types [e.g. pennate diatoms overlaying rounded diatoms (such as Coscinodiscus sp.) or diatom fragments], how did the proposed object-based segmentation approach perform for such cases? Was it able to pick out the individual diatoms within the pile, or did it treat the overlaid diatoms as a single agglomerated entity?

We did not perform a systematic test of the effect of diatom density on the preparations since agglomerations of diatoms were sparse. Rounded (i.e. centric) diatoms were present in only very low numbers, so that the case proposed by the reviewer does not play a significant role in our investigation. However, we observed that individual diatom objects in aggregates were mostly recognized as long as they were still visually separated from each other, which in the agglomerates we observed in our material is true for a substantial number of cells. We addressed this in the discussion in lines 420ff. However, the influence of material density on diatom segmentation is subject to currently ongoing research and will be analysed in a future publication.

3. The authors described their developed object-based tiling methods coupled with an object integrity constraint. However, will this influence the stride of the tiles as well, especially if the diatoms are irregularly spaced apart (as in this strew) and how does the algorithm mitigate this effect, since some tiles may contain (for instance) half of a diatom and the next window may contain the other half of the diatom?

This is exactly the point of the proposed improvements: instead of tiling with a regular stride and thereby often cutting individual objects into pieces, each tile fed into the network during training is positioned around an object; neighbouring objects still present are either kept or masked out, depending on whether they are being substantially cut by the new boundary of the tile. This is also stated in the manuscript on lines 104ff, 193ff and 531ff.

4. The authors mentioned about the splitting of the image subsections into 512*512 tiles in the manuscript, but did not mention how this was performed (e.g. using MATLAB, Python, etc). Please include this information in the Methods section as well.

This is now specified in the Methods section (lines 185, 192 and 206), along with clarifying the subsequent processing (line 236).

5. As this is a study focused on deep learning, please include information such as the training time (per iteration or epoch) required for training the Mask-RCNN & U-Net models, as well as the loss function plots for each of these frameworks during the training. The former would be useful for gauging the relative computational complexity of the frameworks used, while the latter may be used to identify if the model training proceeded optimally.

We did not consider it useful to present absolute training times since they are highly dependent on hardware used, but as a rough comparison, training of the used Mask-RCNN models was about 10 times faster than for the U-Net models (as stated in line 540). 

We fully understand the reviewer’s demand for more information about training performance. As for the loss plots of the 24 training runs performed, we do not believe it would help a reader if we provided all of them graphically, even if as a supplement, but our rationale for selecting the number of training epochs to avoid overfitting is presented in lines 260-269. As stated in the manuscript (line 263), the AP metrics of the mask R-CNN models converged well before training end and did not show overfitting. The loss of the U-Net models converged within the 100 epochs we trained the models (now stated in line 267f).

6. Please include the registered mark (®) after the word NVIDIA (when referring to the V100 GPU) on page 12.

Done.

On a separate note, it would also be good if the authors could include a section on the future potential applications of their current algorithm. Some aspects which the authors might consider to include in this section are as follows:

7. Can this algorithm be used to characterize living diatoms, rather than just the siliceous tests? If unsure, the authors may want to include this point as a statement for future expansion of the current work in optical microscopy.

We see no reason why the proposed methods could not be used for other types of diatom preparations or completely different types of objects; we have now included a statement on this in the Conclusions (line 537f).

8. Can the algorithm be developed further to phylogenetically classify the diatoms based on their frustule or poroid composition (down to their individual species)?

This is an interesting question. For taxonomic identification of diatoms, numerous studies (including one from our group, cited in the paper) have previously tested deep learning methods, though not segmentation architectures. We now added an additional sentence and some references as the last sentence of the Conclusions section (line 547ff).

Reviewer #2: From a theoretical point of view this new automatic technic of diatom analysis seems promising. "Deep learning" in principle is a milestone in the field of object detection and comparison. But some questions have to be answered:

- The author investigated diatoms from the Menne river comprising more than 110 different species. Please provide a brief introduction about those species. Are there any differences in recall and precision among the species.

Unfortunately, we have no information about precision and recall per taxon. The estimate of species richness presented is based on identifying only a subset of ca. 50% of all specimens at the species level; for the presented experiments, only a labelling as “diatom” was used. This is now clarified on lines 111ff.

- There is little information on the diversity of diatoms analyzed. A model that has been trained from images with similarly shaped diatoms will have more difficulty in detecting other diatom shapes.

This is indeed a concern, and for future applications, segmentation models would ideally need to be trained on training sets that are taxonomically representative of the samples to which the segmentation model will be applied. We now added such a statement to the Conclusions (line 541ff) and added Figure 1 comparing the 10 most abundant species to demonstrate the morphological diversity of the diatoms (referenced that in line 113ff).

- How to ensure the precision of human annotator?

Every annotation was seen and checked by at least two different persons to ensure the annotated object was a relevant diatom and that its outline was captured accurately, this is now stated in lines 150ff.

- I suggest that these minor corrections should be done before the publication

Reviewer #3: This paper described an implementation of deploying deep learning-based instance segmentation algorithm, Mask-RCNN, in the area of microorganism detection. It demonstrates that, being well trained on the dataset, Mask-RCNN/U-net have a good performance on recognizing a variety of components and thus can facilitate the analysis of microorganism through image processing. Such applications in biology are not rare (such as Ruiz-Santaquiteria, J., Bueno, G., Deniz, O., Vallez, N., & Cristobal, G. (2020). Semantic versus instance segmentation in microscopic algae detection. Engineering Applications of Artificial Intelligence, 87, 103271).

The authors recommend 3 data preparation/augmentation methods for algorithm training purpose, i.e., 1) fixed-size tiling, 2) object-based positioning, and 3) object-based positioning + object integrity constraint. It was founded that the last method had the best performance. I think the main novelty that the authors intend to present in this paper is the step-by-step improvement among the 3 data augmentation methods. However, to my point of view, the development of these 3 methods is the procedure of eliminating the mistakes made during the process and the improvement in the model performance is no surprise. For example, when the raw image is cut into sub-images, the cutting off objects (below 75%) at the cutting boundary would be wrongly labelled. Expect that the author can prove that this is a routine procedure by the peers, otherwise this error should not have happened during the data preparation and getting rid of such wrongly labelled data should be a normal job (as introduced by the authors in the following 2 methods).

Therefore, my opinion is that the novelty in the paper is not strong enough to convince me to accept it.

If we interpret this comment correctly, then the reviewer suggests that the object-based positioning and object integrity constraint are routinely applied in (diatom?) segmentation analyses. But from our point of view and according to our experience and literature review, this is not the case. In our search through related publications, we have neither been able to find an example (whether for diatoms or for other objects), nor have we been able to find a study systematically investigating their effect, as we did in our manuscript. 

In most diatom applications of computer vision and deep convolutional networks so far, images were taken manually by a diatom expert, i.e., small images that were always centred on and cropped to a single target specimen. In such cases, the target object is naturally left intact and one could consider this equivalent to an implicit object based positioning plus eliminating neighbouring cells by cropping the image so that it contains only a single object. Nevertheless, this approach is not applicable for automated imaging and high throughput.

In the publication cited by the reviewer, complete field-of-view images containing multiple cells plus some debris were analysed. The problem of cropping objects by tiling was not addressed and mitigated only by scaling down the quite large images by 96% (2592x1944 to 480x360 pixel) to fit into the model input size (probably scaled down by 84% for the Mask R-CNN models, but on this no clear details are given in the publication). As a consequence, the resulting segmentation masks will then lack in resolution detail. This is now stated in line 80ff and referred to in lines 178 and 520.

Our manuscript addresses a high throughput situation in which images are captured in an automated manner, by scanning a large contiguous slide area without manual intervention (apart from initially specifying the area to be scanned), in which case object based tile positioning is neither implicit nor is it applied routinely. Also, we segment at the original detail resolution which is essential e.g. for shape-based morphometric analyses and potentially improves masking out irrelevant objects, which might be beneficial for subsequent classification steps (included now in the conclusions section, line 522f).

Reviewer #4: The research work was lacking originality. The techniques used are well-known, and the contribution made by the use of object integrity constraints is minor and not novel. 

As mentioned, we have so far not been able to find previous publications employing this constraint; it would have been helpful if the reviewer had had provided specific references. We are quite certain that object-based positioning and the object integrity constraint have not been applied previously in the context of diatom analyses or virtual slides of other microscopic organisms, but we have not been able to find examples of their use to other types of objects either. Also we are not aware of a previous systematic evaluation of the effects of these approaches.

It's unclear how the object integrity constraint is calculated without first segmenting the object, which appears to be a redundant process.

We would like to underline here again, as also specified in the manuscript, that the object integrity constraint is only used during supervised training of segmentation models, i.e., when previously annotated object outlines are available. At inference time, i.e., when segmenting images, neither object based positioning nor the object integrity constraint plays a role.

The database employed is quite small.

Unfortunately, the reviewer did not mention what size of a database she / he would consider of sufficient size. Nevertheless, we do not agree, that the database is too small to compute significant results. For comparison, in the publication cited by reviewer 3 above (Ruiz-Santaquiteria et al. 2020, one of the recent examples of DL-based segmentation of diatom images), 126 images of 2,592 × 1,944 pixels (= ca. 634 Megapixels) depicting 1,446 specimens belonging to 10 taxa were used in total (training and validation, where it seems validation data was also re-used for evaluation). In our case, the corresponding numbers are six slide scan subsections of 55,000 x 5,000 - 39,000 (varying depending on the sample density) pixels (= in total ca. 4,100 megapixel for training, validation and evaluation data), depicting ca. 3,000 specimens belonging to ca. 110 taxa. We now included this information as the first sentence of the Results section (line 304ff).

---

## [Decision Letter · Decision Letter 1]

10 Oct 2022

PONE-D-22-19514R1Improving deep learning-based segmentation of diatoms in gigapixel-sized virtual slides by object-based tile positioning and object integrity constraintPLOS ONE

Dear Dr. Kloster,

Thank you for submitting your manuscript to PLOS ONE. After careful consideration, we feel that it has merit but does not fully meet PLOS ONE’s publication criteria as it currently stands. Therefore, we invite you to submit a revised version of the manuscript that addresses the points raised during the review process. Please revise the manuscript based on the reviewers suggestions and comments  

We look forward to receiving your revised manuscript.

Kind regards,

Kathiravan Srinivasan

Academic Editor

PLOS ONE

Reviewers' comments:

Reviewer's Responses to Questions

**Comments to the Author**

1. If the authors have adequately addressed your comments raised in a previous round of review and you feel that this manuscript is now acceptable for publication, you may indicate that here to bypass the “Comments to the Author” section, enter your conflict of interest statement in the “Confidential to Editor” section, and submit your "Accept" recommendation.

Reviewer #1: All comments have been addressed

Reviewer #2: All comments have been addressed

Reviewer #3: (No Response)

2. Is the manuscript technically sound, and do the data support the conclusions?

Reviewer #1: Yes

Reviewer #2: Yes

Reviewer #3: Partly

3. Has the statistical analysis been performed appropriately and rigorously? 

Reviewer #1: Yes

Reviewer #2: Yes

Reviewer #3: Yes

4. Have the authors made all data underlying the findings in their manuscript fully available?

Reviewer #1: Yes

Reviewer #2: Yes

Reviewer #3: Yes

5. Is the manuscript presented in an intelligible fashion and written in standard English?

Reviewer #1: Yes

Reviewer #2: Yes

Reviewer #3: Yes

6. Review Comments to the Author

Reviewer #1: 1. The authors have addressed most of the comments as provided for the previous manuscript submission. However, for Reviewer #1 Comment #1, the authors responded that they determined the area of the diatom covered by the tile by counting the number of pixels and comparing this with the area of the diatom (also by enumerating the number of pixels covered by the diatom). Nonetheless, they did not clearly indicate this approach in their manuscript (especially in lines 193-206, where they supposedly describe the tiling and segmentation of the diatoms). It would thus be prudent to mention this method in the manuscript to avoid confusion by the reader. In the same regard, the authors may also need to address the issue of the diatoms spanning across multiple focal planes since not all diatoms would be expected to reside in the same focal plane - did they acquire an extended depth-of-field (EDF) image of the diatom and compute the area of the diatom by counting the number of pixels occupied by the diatom (assuming that it was residing in a single 2D plane, instead of spanning across a 3D volume)? If so, wouldn't this be subject to inaccuracies, since the actual area covered by the diatom would actually be greater than what was being computed in this context?

2. Line 256 seems to indicate some issue with the double quotation marks - please assist to resolve this matter.

3. Line 290 needs to be rewritten as "The output matrix for each input pixel contained the prediction score for a sample belonging to the diatom class".

Reviewer #2: (No Response)

Reviewer #3: After checking the authors' response, I still insist my decesion on my 1st revision, which is rejection.

7. PLOS authors have the option to publish the peer review history of their article (what does this mean?). If published, this will include your full peer review and any attached files.

Reviewer #1: No

Reviewer #2: **Yes: **Jian Zhao

Reviewer #3: No

---

## [Author Response · Author response to Decision Letter 1]

17 Oct 2022

PONE-D-22-19514R2

Improving deep learning-based segmentation of diatoms in gigapixel-sized virtual slides by object-based tile positioning and object integrity constraint Dr. Michael Kloster

Dear Dr. Kloster,

We've checked your submission and before we can proceed, we need you to address the following issues:

1. Thank you for stating the following financial disclosure:

"This work was funded by the Deutsche Forschungsgemeinschaft (DFG) in the framework of the priority programme SPP 1991 Taxon-OMICS under grant nrs. BE4316/7-1 & NA 731/9-1. D.L.'s contribution was supported by the German Federal Ministry for Economic Affairs and Energy (BMWi) (FKZ: 0324254D). BIIGLE is supported by the BMBF-funded de.NBI Cloud within the German Network for Bioinformatics Infrastructure (de.NBI) (031A537B, 031A533A, 031A538A, 031A533B, 031A535A, 031A537C, 031A534A, 031A532B)." 

The statement is included in the cover letter now.

2. Please ensure that you refer/cite to Table 1-4 in your text as, if accepted, production will need this reference to link the reader to the Table.

All table, figures and supplemental information are referenced using the citation system integrated in MS-WORD. A (minor) problem we have with this approach is that we did not find a way to address multiple figures in the way described in the authoring guidelines (e.g. Figs 3 and 4, Figs 5-7) and that the bold font formatting from the figure / table caption is also used on the reference links.

Larger files of the supplemental information are available via Zenodo, but we did not upload some figures / tables yet to any repository and thus included them in the manuscript. Please drop us a note if we are supposed to do so, from my understanding there will be uploaded to FigShare by PLOS ONE.

---

## [Decision Letter · Decision Letter 2]

24 Oct 2022

Improving deep learning-based segmentation of diatoms in gigapixel-sized virtual slides by object-based tile positioning and object integrity constraint

PONE-D-22-19514R2

Dear Dr. Kloster,

We’re pleased to inform you that your manuscript has been judged scientifically suitable for publication and will be formally accepted for publication once it meets all outstanding technical requirements.

Kind regards,

Kathiravan Srinivasan

Academic Editor

PLOS ONE

Additional Editor Comments (optional):

Reviewers' comments:

Reviewer's Responses to Questions

**Comments to the Author**

1. If the authors have adequately addressed your comments raised in a previous round of review and you feel that this manuscript is now acceptable for publication, you may indicate that here to bypass the “Comments to the Author” section, enter your conflict of interest statement in the “Confidential to Editor” section, and submit your "Accept" recommendation.

Reviewer #1: All comments have been addressed

Reviewer #2: All comments have been addressed

2. Is the manuscript technically sound, and do the data support the conclusions?

Reviewer #1: Yes

Reviewer #2: Yes

3. Has the statistical analysis been performed appropriately and rigorously? 

Reviewer #1: Yes

Reviewer #2: Yes

4. Have the authors made all data underlying the findings in their manuscript fully available?

Reviewer #1: Yes

Reviewer #2: Yes

5. Is the manuscript presented in an intelligible fashion and written in standard English?

Reviewer #1: Yes

Reviewer #2: Yes

6. Review Comments to the Author

Reviewer #1: The current manuscript presents a viable DNN-mediated approach to object segmentation in micrographs of diatoms, which poses significant importance in the field of marine ecology and biodiversity. The methods utilized in the study are also well described & detailed, and the authors recognize the limitations of their current approach (e.g. the use of focus stacking to enhance the depth-of-field) may present obfuscated results in segmentation. However, they mentioned that they still chose to implement this approach to allow comparison of the Mask R-CNN models with U-Net models (the latter only being unable to separate overlapping objects).

The manuscript may have to be reviewed again for spelling and grammatical errors though, as some typographical errors (such as 'und' which should be written as 'and', line 383) seem to have gone amiss in this version.

Reviewer #2: All comments have been addresed. The artical have been well revised. Thanks for your wonderful work.

7. PLOS authors have the option to publish the peer review history of their article (what does this mean?). If published, this will include your full peer review and any attached files.

Reviewer #1: No

Reviewer #2: No
